# Auditing Test Data Contamination with Error Rate Control for Reliable LLM Evaluation

## Abstract

Large language models (LLMs) have achieved impressive performance across diverse tasks, largely driven by large-scale pretraining data. However, this data abundance has led to a critical issue: test data contamination, where benchmark datasets inadvertently overlap with pretraining corpora. This contamination compromises the reliability of LLM evaluation by making it difficult to distinguish genuine generalization from memorization. To address this challenge, existing training data detectors aim to identify clean (unseen) data within potentially contaminated test sets. While effective to some extent, these methods often misclassify contaminated data as clean due to the black-box nature of LLMs, resulting in residual contamination and unreliable evaluation. This raises a key question: Can we control the proportion of contaminated data mistakenly identified as clean i.e., false discovery rate (FDR), below a user-specified threshold, while maximizing the amount of clean data retained for evaluation? Thus, we propose TD4Eval, a principled framework for training data detection that simultaneously ensures strict FDR control and high detection power. Specifically, we propose a rejection-count-based adaptive weighting strategy that learns the relative contribution of each detector. Based on these weights, we integrate multiple complementary detectors and apply the Benjamini-Hochberg (BH) procedure to control the FDR. Theoretically, we show that TD4Eval achieves asymptotic optimality in controlling FDR and maintaining high power. Empirical results on real-world datasets demonstrate that TD4Eval achieves an average 30% improvement in FDR over SOTA methods.

## 1 Introduction

Large language models (LLMs) have demonstrated remarkable capabilities across a wide range of domains, largely due to the massive volumes of data used during pretraining Cheng et al. (2025); Chang et al. (2024). However, despite these impressive achievements, the uncontrolled expansion of data has introduced the problem of data contamination, where test datasets are inadvertently included in the training corpus. This contamination makes it difficult to determine whether a model has genuinely acquired a capability or is simply memorizing test data, thereby undermining the reliability of evaluation Zhang et al. (2024c); Lv et al. (2024); Yao et al. (2024); Sun et al..

To address the issue of test data contamination, a variety of training data detection methods have been proposed Shi et al.; Zhang et al. (2024b;a); Golchin & Surdeanu (2023); Jacovi et al.. These approaches typically frame the task as a binary classification problem, aiming to distinguish between contaminated data (i.e., data that has been seen by large language models during training) and clean data (i.e., data not present in the pre-training set) within a potentially contaminated dataset. The identified clean data can then be used for reliable evaluation of LLMs, mitigating the impact of contamination on LLM evaluation.

However, due to the black-box nature of LLMs, even state-of-the-art detection models struggle to perfectly separate contaminated from clean data. That means a significant proportion of contaminated samples may be misclassified as clean, i.e., false positives, leading to residual contamination in the selected evaluation set and undermining the reliability of model evaluation (as demonstrated in the Experiment section 5.4). If the proportion of false positives can be controlled below a user-specified threshold, residual contamination can be effectively reduced. From the perspective of statistical hypothesis testing, this problem can be formulated as a false discovery rate (FDR) control

problem. Classical approaches, such as the Benjamini—Hochberg (BH) procedure Benjamini & Hochberg (1995); Ferreira & Zwinderman (2006), can be employed to achieve this control. While these FDR-controlled methods exhibit desirable statistical properties Benjamini & Hochberg (1995); Benjamini & Yekutieli (2001), they often overlook another critical objective in training data detection: maximizing the amount of clean data identified, also known as detection power. Insufficient detection power may result in only a small number of clean samples being retained, thereby compromising the representativeness and robustness of subsequent evaluation. This challenge motivates a key question: Can we achieve the FDR control while maximizing detection power for training data detection?

To settle this problem, we propose TD4Eval, a training data detection framework designed for reliable LLM evaluation, with a dual focus on FDR control and power maximization. Specifically, we introduce a novel fusion strategy that integrates multiple detection techniques, such as PPL Li (2023), Min-k Shi et al., and Min-k++ Zhang et al. (2024b), into the classical BH. Our approach proceeds in three steps: first, test statistics are extracted from each detection method for every candidate sample to compute individual p-values. Then, due to data heterogeneity, i.e., detectors may exhibit different performance across datasets, we propose a rejection-count-based adaptive weighting strategy that adaptively learns the relative contribution of each detector. Based on these weights, we integrate multiple complementary detectors through the weighted Cauchy combination. Finally, we apply the BH procedure to control the overall FDR with power maximization.

From a theoretical perspective, due to data heterogeneity, we adopt a data-driven weighting scheme instead of fixed weights. However, since these weights are estimated from the same sample that generates the p-values, they become stochastically dependent on the underlying test statistics, rendering previous theoretical results based on fixed weights inapplicable (Bates et al., 2023; Wu et al., 2023; Long et al., 2023). To address this challenge, we estabilish the convergence of data-driven weights and extend the analysis for converged weights. The new theoretical results show TD4Eval achieves asymptotic optimality in controlling FDR and maintaining high power.

Finally, we validate the effectiveness of TD4Eval through experiments on both real-world datasets and practical evaluation benchmark. Specifically, we conduct experiments on real datasets WikiMIAShi et al., arXivTection Duarte et al. (2024), BBC Real-Time Li et al. (2024), and MIMIR (Duan et al.), and find that TD4Eval achieves an average improvement of 30% in FDR performance compared to the SOTA, demonstrating its effectiveness in controlling false positives. In the real-world benchmark evaluation, we construct a contaminated setting by fine-tuning LMMs on the 4 classical LLM evaluation benchmarks. The results demonstrate that TD4Eval successfully identifies clean data with a controlled FDR, effectively mitigating the impact of data contamination and preserving the reliability of the model evaluation.

- To the best of our knowledge, we are the first to propose controlling the FDR while maximizing power for detecting clean data under data contamination. This effectively mitigates the impact of contaminated data and enhances the reliability of the model evaluation.
- We introduce TD4Eval, a training data detection method that offers theoretical guarantees for FDR control and power maximization.
- We validate the effectiveness of our approach on real-world datasets and demonstrate its capability to support reliable model evaluation.

## 2 RELATED WORKS

**Data Contamination.** Data contamination has been extensively studied in the literature Mann et al. (2020); Magar & Schwartz (2022); Deng et al. (2024); Golchin & Surdeanu (2024), where training data may inadvertently include evaluation benchmark data, leading to unreliable evaluation results. As such, assessing the potential leakage of benchmark data into pretraining corpora is essential for trustworthy model evaluation Dong et al. (2024); Dekoninck et al. (2024); Wang et al.; Jain et al.; Oren et al. (2023); Golchin & Surdeanu. In fact, the problem of data contamination can, to some extent, be viewed as a specific instance of Membership Inference Attacks (MIA) Das et al. (2024); Duan et al. (2024), which aim to determine whether a given data point was part of a model's training set. Motivated by this connection, several recent works have approached training data detection in LLMs from the perspective of MIA. For example, Shi et al. hypothesize that clean data is more likely to contain outlier tokens that result in significantly higher loss values. Based on this observation, they propose the Min-K% method, which identifies contaminated data by analyzing the top-k token

log probabilities. Building on this, Zhang et al. (2024b) introduce Min-K%++, grounded in the theoretical insight that training samples tend to correspond to local maxima of the model's likelihood function along each input dimension. Furthermore, Zhang et al. (2024a) observe that after fine-tuning the model with a small set of unseen data, the perplexity of LLMs shifts differently for contaminated versus clean data. Leveraging this behavior, they propose the FSD method.

**FDR Control.** Reformulating the training data detection problem as a multiple testing problem, the proportion of false positives among the clean data corresponds to the false discovery rate (FDR). To ensure that the FDR remains below a user-specified threshold, extensive research in the statistical literature has focused on FDR control. It serves as a powerful tool for reliably identifying true positives while limiting excessive false discoveries Benjamini & Hochberg (1995); Benjamini & Yekutieli (2001), which is crucial in a wide range of application domains, such as genomics Storey & Tibshirani (2003) and healthcare Genovese et al. (2002). For example, Zhang et al. (2022b) investigate strategies to improve model selection under FDR control, while Bates et al. (2023) introduce a method to control the FDR for a fixed novelty detection model. However, their work does not consider the theoretical guarantees under statistical power and data-driven weights.

In contrast to previous work, we argue that simultaneously controlling the FDR and maximizing detection power on clean data is critical for training data detection, as residual contamination can undermine the reliability of model evaluations. Therefore, we propose TD4Eval, designed to achieve rigorous and reliable evaluations of LLMs. In addition, we establish theoretical guarantees for both FDR control and statistical power, highlighting the theoretical validity of our method.

## 3 PROBLEM DEFINITIONS

Suppose we are given a dataset in which a portion of the data is contaminated. The dataset consists of $n$ samples, denoted by $Z_1, Z_2, \ldots, Z_n$, and indexed by $\mathcal{D}_{\text{total}} = [n] = \{1, 2, \ldots, n\}$. Each sample belongs either to the contaminated data or the clean data. Specifically, if a data sample has been seen during the training process of a LLM, it is considered **contaminated**; otherwise, it is considered **clean**. Accordingly, we define two disjoint index sets: $\mathcal{D}_{\text{con}} \subseteq \mathcal{D}_{\text{total}}$: indices of contaminated data, $\mathcal{D}_{\text{clean}} = \mathcal{D}_{\text{total}} \setminus \mathcal{D}_{\text{con}}$: indices of clean data.

The goal of training data detection is to estimate the set of clean data, denoted by $\hat{\mathcal{S}} \subseteq \mathcal{D}_{\text{total}}$, which can be used to support reliable LLM evaluation. This task can be formulated as a classification problem: determining whether a given sample $Z_i$ belongs to the $\mathcal{D}_{\text{con}}$ or $\mathcal{D}_{\text{clean}}$. The decision is based on a detection score $S_i$, where, in general, lower scores are assumed to indicate a higher likelihood that the sample was seen during training. Formally, the prediction function is defined as:

$$h(Z_i) = \begin{cases} \text{contaminated data} & \text{if } S_i < t, \\ \text{clean data} & \text{if } S_i \geq t, \end{cases} \tag{1}$$

where $t$ is a threshold determined based on the distribution of detection scores.

To compute such scores, many methods have been proposed, such as Min-K%, which uses the average probability of the $k\%$ outlier tokens with the lowest predicted probabilities. While these methods can distinguish clean and contaminated data to some extent, the black-box nature of LLMs, including the inaccessibility of their full pretraining data and model architecture, makes this task particularly challenging. As a result, even SOTA detection methods often yield clean data that contain a significant proportion of contaminated data, thereby undermining the reliability of evaluation.

In this paper, we adopt a statistical perspective and aim to control the proportion of contaminated data in selected set $\hat{\mathcal{S}}$. Specifically, for each sample $i \in \mathcal{D}_{\text{total}}$, we consider the following hypothesis test:

$$\begin{cases} \mathbb{H}_0 : \text{Sample } i \text{ is from contaminated data}, \\ \mathbb{H}_1 : \text{Sample } i \text{ is from clean data} \end{cases} \tag{2}$$

Our objective is to control the FDR of $\hat{\mathcal{S}}$ to be below a user-specified threshold $\alpha$, while simultaneously maximizing the power of the detection. These quantities are formally defined as:

$$\text{FDR} := \mathbb{E}\left[\frac{|\hat{\mathcal{S}} \cap \mathcal{D}_{\text{con}}|}{|\hat{\mathcal{S}}| \vee 1}\right] \quad \text{and} \quad \text{Power} := \mathbb{E}\left[\frac{|\hat{\mathcal{S}} \cap \mathcal{D}_{\text{clean}}|}{|\mathcal{D}_{\text{clean}}| \vee 1}\right]. \tag{3}$$

# 4 METHODS

In this section, we introduce TD4Eval, a method for detecting non-training data with rigorous FDR control and power maximization, enabling reliable evaluation of LLMs. Specifically, we first formalize the training data detection task as a multiple hypothesis testing problem and review the classical Benjamini-–Hochberg (BH) Benjamini & Hochberg (1995) procedure for FDR control. Building upon BH, we then propose a cauchy fusion that integrates existing training data detection methods to maximize detection power. Finally, we provide theoretical guarantees for the TD4Eval.

## 4.1 BH PROCEDURE

To control the FDR, we formulate the training data detection task as a multiple hypothesis testing problem. Specifically, for each test sample $Z_i$, we test the null hypothesis,

$$H_0^{(i)} : \text{"Sample } Z_i \text{ is from the contaminated data".}$$

By computing a p-value $p_i$ for each sample, we obtain a collection of $n$ hypotheses $\{H_0^{(i)}\}_{i=1}^n$, one for each test sample. This formulation enables the application of classical multiple testing procedures to determine which hypotheses to reject.

To obtain the p-value, we first compute a detection score $S_i$ for each sample $Z_i$ using a given detection method. Let $\mathcal{S}_{\text{ref}}$ denote the contaminated data detection score distribution. Although the true distribution of contaminated data is not directly accessible, it can be approximated by collecting samples from commonly used pretraining sources, such as Wikipedia, that are from the same domain and were published prior to the model's release date. For example, WIKIMIA Shi et al. considers Wikipedia articles created before 2017 as contaminated data, because many pretrained models, including LLaMA and GPT-NeoX, were released after 2017 and incorporate Wikipedia dumps into their pretraining corpora. The p-value for sample $Z_i$ is then defined as:

$$p_i = \mathbb{P}_{S \sim \mathcal{S}_{\text{ref}}}(S \leq S_i). \tag{4}$$

BH procedure is a classical method in multiple hypothesis testing, which aims to control the error rate by identifying a data-dependent threshold such that FDR is bounded by a target level $\alpha \in (0, 1)$. Specifically, given a set of p-values $\{p_i\}_{i=1}^n$, we first sort them in ascending order: $p_{(1)} \leq p_{(2)} \leq \cdots \leq p_{(n)}$, and then find the largest index:

$$m = \max\left\{ j \in [n] : p_{(j)} \leq \frac{j\alpha}{n} \right\}, \tag{5}$$

The resulting BH threshold is then given by $t_{\text{BH}} = \frac{m\alpha}{n}$. Finally, all p-values smaller than $t_{\text{BH}}$ are considered statistically significant, leading to the rejection of the null hypothesis $H_0$, and the corresponding samples are identified as clean data. The final $\widehat{\mathcal{S}}$ can be represent as:

$$\widehat{\mathcal{S}} = \{Z_j \in \mathcal{D}_{\text{total}} : p_j < t_{\text{BH}}\}. \tag{6}$$

## 4.2 TD4EVAL

While the BH procedure effectively controls the FDR, it overlooks the goal of maximizing statistical power. This can result in only a small number of clean samples being detected, thereby compromising the comprehensiveness of model evaluation. Recent advances in training data detection have introduced various scoring methods (e.g., Min-K Shi et al., Min-K++ Zhang et al. (2024b), etc.), each capturing different aspects of the data. These methods provide complementary information, which motivates the idea of combining multiple statistics to construct a more powerful p-value. The framework is shown in Figure 1.

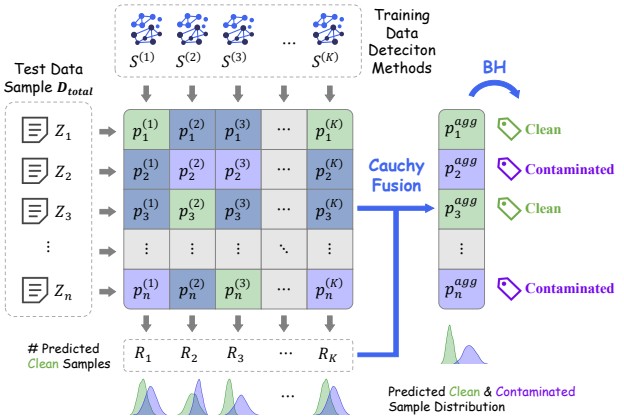

Figure 1: The framework of TD4Eval.

Suppose we have a set of training data detection methods, each associated with a score function $S^{(1)}, \ldots, S^{(K)}$. The corresponding p-values based on these score functions are denoted as $\{p_j^{(1)}\}_{j \in [n]}, \ldots, \{p_j^{(K)}\}_{j \in [n]}$, respectively. Since the performance of each detection method may vary, we aim to better integrate the information from different models by assigning them appropriate weights. Intuitively, the more non-training data points a method successfully identifies (i.e., rejects the null hypothesis $H_0$), the more it contributes to the overall power (Zhang et al., 2022b). Therefore, we compute the weight of each method $S^{(k)}$ based on the number of rejections it makes under a controlled FDR:

$$R_k = \left| \left\{ Z_i \in \mathcal{D}_{\text{total}} : p_i^{(k)} \leq t_{\text{BH}}^{(k)} \right\} \right|, \quad w_k = \frac{R_k}{\sum_{j=1}^{K} R_j}. \tag{7}$$

Here, $t_{\text{BH}}^{(k)}$ is the rejection threshold determined by the BH procedure over $\{p_j^{(k)}\}_{j \in [n]}$. Inspired by Liu & Xie (2020), we map the individual p-value to the Cauchy space and compute a weighted sum:

$$T_i = \sum_{k=1}^{K} w_k \cdot \tan\left[ \left( 0.5 - p_i^{(k)} \right) \pi \right]. \tag{8}$$

The reason why we choose the Cauchy distribution is that it is a heavy-tailed distribution. This implies that when an individual p-value is very small (i.e., highly significant), its corresponding Cauchy-transformed value becomes extremely large (approaching infinity). Such behavior allows the combined statistic $T_i$ to be dominated by the most significant evidence among the individual tests, thereby increasing the sensitivity to strong signals while maintaining robustness under $H_0$.

The combined test statistic $T_i$ approximately follows a standard Cauchy distribution. To map it back to the p-value space, we apply the inverse Cauchy transformation. The resulting aggregated p-value can then be used for FDR control. Specifically, the final p-value is computed as:

$$p_i^{\text{agg}} = \frac{1}{2} - \frac{1}{\pi} \arctan(T_i). \tag{9}$$

The combined p-value $p_i^{\text{agg}}$ can then be used in the BH procedure to control the overall FDR and maximize power. The final threshold $t$ is defined as:

$$t_{final} = \frac{\alpha}{n} \max\left\{ j \in [n] : p_{(j)}^{\text{agg}} \leq \frac{j\alpha}{n} \right\}, \tag{10}$$

where $p_{(j)}^{\text{agg}}$ denotes the $j$th smallest value among the set $\{p_i^{\text{agg}}\}_{i \in [n]}$. The final $\widehat{\mathcal{S}}$ can be represent as:

$$\widehat{\mathcal{S}} = \{Z_j \in \mathcal{D}_{\text{total}} : p_j < t_{final}\}. \tag{11}$$

### 4.3 THEORETICAL GUARANTEE

In this section, we theoretically prove that TD4Eval can effectively control the FDR while maximizing statistical power. We begin by presenting two key lemmas, which form the theoretical foundation for establishing the FDR control guarantee of the TD4Eval procedure. Specifically, we first characterize the statistical properties of the aggregated p-values used in TD4Eval. Based on these properties, we then demonstrate the effectiveness of TD4Eval in controlling the FDR.

**Lemma 1.** *Bates et al. (2023) Under the null hypothesis $H_0$, suppose the score $S_i^{(k)}$ is drawn from the same distribution as the reference data, i.e., $S_i^{(k)} \sim \mathcal{S}_{ref}^{(k)}$, and the p-value is defined as $p_i^{(k)} = \mathbb{P}_{S \sim \mathcal{S}_{ref}^{(k)}}(S \leq S_i^{(k)})$. Then, under $H_0$, the p-value $p_i^{(k)}$ follows a uniform distribution: $p_i^{(k)} \sim \text{Uniform}[0, 1]$.*

Lemma 1 has been proved by Bates et al. (2023). Building on Lemma 1, consider the p-values $p_i^{(1)}, \ldots, p_i^{(K)}$ computed from different training data detectors. Each of these p-values satisfies,

$$p_i^{(k)} \sim \text{Uniform}[0, 1] \quad \text{under the null hypothesis } H_0.$$

Then, the Cauchy-aggregated p-values $\{p_i^{\text{agg}}\}_{i \in [n]}$ can be approximately uniformly distributed under fixed weights (Liu & Xie, 2020), and this result has been extended to various structures (Wu et al.,

2023; Long et al., 2023). However, as we use a rejection-count-based adaptive weighting strategy that adaptively learns the relative contribution of each detector, these data-driven weights introduce additional theoretical challenges against fixed weights. Specifically, the estimated weights are computed from the same sample that generates the p-values, so they are stochastically dependent on the underlying test statistics. To address this issue, we first establish Lemma 2, which demonstrates that the data-driven weights converge to fixed constants. We then present Lemma 3, which ensures that the aggregated p-values preserve uniformity.

**Lemma 2** (Convergence of the weights for TD4Eval). *Consider $K$ detection methods, each with score function $S^{(1)}, \ldots, S^{(K)}$. For each method $k$, let p-values on the total dataset $\mathcal{D}_{\mathrm{total}}$ be $\{p_i^{(k)}\}_{i=1}^n$. Apply the BH procedure at level $\alpha$ separately to each method, producing the rejection threshold $t_{\mathrm{BH}}^{(k)}$ and the number of rejections $R_k = \left| \left\{ Z_i \in \mathcal{D}_{\mathrm{total}} : p_i^{(k)} \leq t_{\mathrm{BH}}^{(k)} \right\} \right|$. Define the normalized data-driven weights $w_k = \frac{R_k}{\sum_{j=1}^K R_j}$. Then, there exist deterministic constants $(w_1^*, \ldots, w_K^*)$ with $\sum_{k=1}^K w_k^* = 1$ such that $w_k \xrightarrow{\text{a.s.}} w_k^*$ for each $k = 1, \ldots, K$.*

**Lemma 3** (Uniformity of aggregated p-values in TD4Eval). *Let $p_i^{(1)}, \ldots, p_i^{(K)}$ be the p-values corresponding to a given sample $i$, each satisfying $p_i^{(k)} \sim \mathrm{Uniform}[0, 1]$ under the null hypothesis $H_0$. We have that the aggregated p-value $p_i^{\mathrm{agg}}$ is uniformly distributed on $[0, 1]$ under $H_0$.*

**Remark 1.** *A detailed theoretical analysis supporting Lemmas 2 and 3 is provided in Appendix 9 . In addition, we empirically evaluate the distribution of aggregated p-values $p_i^{\mathrm{agg}}$ under the null hypothesis on real datasets. The results show that the aggregated p-values are uniformly distributed on $[0, 1]$, which provides empirical evidence supporting the validity of Lemma 3.*

Based on the result of Lemma 3, we have the following theorems:

**Theorem 1** (FDR Control of TD4Eval). *Suppose we are given a set of training data detection methods, each associated with a score function $S^{(1)}, \ldots, S^{(K)}$. Let the corresponding p-values be defined as $\{p_j^{(1)}\}_{j \in [n]}, \ldots, \{p_j^{(K)}\}_{j \in [n]}$, where each p-value is computed as $p_i^{(k)} = \mathbb{P}_{S \sim \mathcal{S}_{ref}^{(k)}}(S \leq S_i^{(k)})$. Then, the TD4Eval procedure controls the FDR at level $\alpha$, i.e.,*

$$FDR := \mathbb{E}\left[ \frac{|\hat{\mathcal{S}} \cap \mathcal{D}_{con}|}{|\hat{\mathcal{S}}| \vee 1} \right] \leq \alpha. \tag{12}$$

**Theorem 2** (Asymptotic Power Consistency of TD4Eval). *Assume that for each $k \in [K]$ and $Z_j \in \mathcal{D}_{\mathrm{clean}}$, $\Pr(p_j^{(k)} \leq c) \geq 1 - \delta$ where $\delta = o(\sqrt{\log n / n})$ and $c$ is a constant, and the density function of $p_j^{\mathrm{agg}}$ for $Z_j \in \mathcal{D}_{\mathrm{clean}}$ has an upper bound $C_f > 0$, we have,*

$$Power := \mathbb{E}\left[ \frac{|\hat{\mathcal{S}} \cap \mathcal{D}_{clean}|}{|\mathcal{D}_{clean}| \vee 1} \right] \geq 1 - C\sqrt{\log n / n}. \tag{13}$$

The assumption in Theorem 2 is standard for proving power consistency (Genovese et al., 2002; Weinstein et al., 2023). Specifically, we show that the power of TD4Eval is lower bounded by $1 - C\sqrt{\log n / n}$, where $C > 0$ is a constant depending on the testing level, the null proportion, and certain distributional characteristics. Consequently, as $n \to \infty$, the power converges to 1. Through Theorems 1 and 2, we demonstrate that TD4Eval achieves asymptotic optimality in controlling the FDR while maintaining high statistical power. Details will be presented in Appendix 10.

## 5 EXPERIMENTS

In this section, we first introduce the baselines, datasets, and experimental setup. We then present the experimental results to demonstrate the effectiveness of our model. The code is available for review [1].

### 5.1 SETUP

**Baselines** We select several baselines from prior work on training data detection, including PPL Li (2023), Lowercase Carlini et al. (2021), Zlib Carlini et al. (2021), Grad Hu et al., Min-K% Shi et al., and Min-K%++ Zhang et al. (2024b). In addition, to assess the effectiveness of our adaptive weighting strategy, we construct three TD4Eval variants: BH-Average, BH-Random, and BH-Max. Detailed descriptions are provided in the Appendix 11.1.

[1]https://anonymous.4open.science/r/TD4Eval-0D60/

Table 1: The detection performance on three datasets. FDR (lower is better), Power (higher is better), ACC (higher is better), AUC (higher is better). Bold indicates the best result per model. For TD4Eval, we report the mean and standard deviation across ten independent runs.

| Method | WikiMIA | | | | | | | | arXivTection | | | | | | | | BBC Real Time | | | | | | | |
|---|---|---|---|---|---|---|---|---|---|---|---|---|---|---|---|---|---|---|---|---|---|---|---|---|
| | Pythia-2.8B | | | | LLaMA-13B | | | | Pythia-2.8B | | | | LLaMA-13B | | | | Pythia-2.8B | | | | LLaMA-13B | | | |
| | FDR↓ | Power↑ | ACC↑ | AUC↑ | FDR↓ | Power↑ | ACC↑ | AUC↑ | FDR↓ | Power↑ | ACC↑ | AUC↑ | FDR↓ | Power↑ | ACC↑ | AUC↑ | FDR↓ | Power↑ | ACC↑ | AUC↑ | FDR↓ | Power↑ | ACC↑ | AUC↑ |
| PPL | 0.141 | 0.980 | 0.913 | 0.983 | 0.145 | 1.000 | 0.919 | 0.999 | 0.149 | 0.909 | 0.873 | 0.945 | 0.138 | 0.960 | 0.901 | 0.975 | 0.156 | **0.802** | 0.825 | 0.882 | 0.137 | 0.931 | 0.890 | 0.950 |
| Lowercase | 0.154 | 0.926 | 0.884 | 0.954 | 0.149 | 0.999 | 0.916 | 0.996 | 0.216 | 0.615 | 0.718 | 0.825 | 0.154 | 0.832 | 0.837 | 0.922 | 0.173 | 0.694 | 0.772 | 0.853 | 0.133 | 0.928 | 0.892 | 0.949 |
| Zlib | 0.178 | 0.768 | 0.809 | 0.880 | 0.150 | 0.959 | 0.899 | 0.977 | 0.200 | 0.580 | 0.712 | 0.822 | 0.160 | 0.819 | 0.829 | 0.909 | 0.207 | 0.550 | 0.700 | 0.797 | 0.151 | 0.832 | 0.840 | 0.903 |
| Grad | 0.174 | 0.798 | 0.823 | 0.907 | 0.154 | 0.929 | 0.885 | 0.964 | 0.150 | 0.798 | 0.825 | 0.907 | 0.139 | 0.913 | 0.881 | 0.881 | 0.188 | 0.645 | 0.745 | 0.827 | 0.151 | 0.853 | 0.849 | 0.921 |
| Min-K% | 0.150 | 0.982 | 0.908 | 0.982 | 0.151 | 1.000 | 0.915 | 0.999 | 0.135 | **0.962** | 0.904 | 0.972 | 0.141 | 0.995 | 0.915 | **0.994** | 0.187 | 0.656 | 0.750 | 0.833 | 0.138 | 0.910 | 0.881 | 0.942 |
| Min-K%++ | 0.145 | 0.997 | 0.918 | **0.993** | 0.150 | 1.000 | 0.915 | **1.000** | 0.148 | 0.906 | 0.872 | 0.948 | 0.137 | 0.995 | 0.917 | 0.991 | 0.167 | 0.744 | 0.795 | 0.866 | 0.133 | 0.935 | 0.894 | 0.950 |
| **TD4Eval** | **0.101** | **0.999** | **0.946** | **0.993** | **0.099** | **1.000** | **0.947** | 0.998 | **0.090** | 0.942 | **0.923** | **0.973** | **0.092** | **0.997** | **0.947** | **0.994** | **0.095** | 0.771 | **0.843** | **0.910** | **0.094** | **0.948** | **0.924** | **0.968** |
| (std) | ±0.018 | ±0.001 | ±0.011 | ±0.001 | ±0.012 | ±0.000 | ±0.007 | ±0.001 | ±0.014 | ±0.010 | ±0.007 | ±0.003 | ±0.020 | ±0.002 | ±0.012 | ±0.002 | ±0.013 | ±0.017 | ±0.003 | ±0.002 | ±0.005 | ±0.002 | ±0.003 | ±0.002 |

**Models and Datasets**   We adopt three LLMs to evaluate our TD4Eval: Pythia-2.8B(Biderman et al., 2023), OPT-6.7B Zhang et al. (2022a) and LLaMA-13B Touvron et al. (2023). Throughout our experiments, we use the checkpoints provided by Hugging Face. As for the evaluation datasets, we employ four benchmark datasets for evaluations, including WikiMIA (Shi et al.), ArXivTection (Duarte et al. (2024)), BBC Real Time (Li et al. (2024)), MIMIR (Duan et al.). Previous works have demonstrated that model developers commonly use text content among those datasets for pre-training (Shi et al.; Duarte et al. (2024)). Detailed information is shown in Appendix 11.2.

**Evaluation metrics**   In this paper, we focus on two key metrics, FDR and Power (Eq. (3)), which play a crucial role in training data detection for LLM evaluation. A lower FDR and a higher Power indicate better detection performance. Additionally, we report the Accuracy (ACC) and AUC, where a higher ACC and AUC reflects more precise detection results.

**Implementation details**   The detailed implementation can be found in in Appendix 11.3.

### 5.2 MAIN RESULT

In this section, We conduct a comprehensive comparison of various training data detection methods across three datasets and multiple language models. Table 1 summarizes the detection performance of all evaluated methods across three datasets and under multiple LLM backbones. Results for OPT-6.7B and MIMIR are provided in the Appendix 11.4. The key findings are as follows: ❶ *TD4Eval achieves the lowest FDR across all datasets and model settings, demonstrating its strong ability to suppress false positives.* Compared to the strongest baseline, TD4Eval achieves a relative FDR reduction of up to 30–39%, indicating a substantial improvement in precision when identifying clean data. ❷ *TD4Eval maintains competitive or superior detection power in most settings.* It achieves near-perfect power on two datasets and performs robustly on the third, showing that it can effectively recall clean samples while maintaining low FDR. ❸*TD4Eval consistently achieves the highest overall accuracy across different datasets and models.* This reflects its comprehensive effectiveness in both minimizing false detections and correctly identifying clean data, which is crucial for downstream applications.

### 5.3 THE ANALYSIS OF TD4EVAL

**Effect of Cauchy Fusion.** We adapt a weighted Cauchy fusion strategy as defined in Eq. (8) and Eq. (9). In this section, we evaluate its effectiveness from two perspectives. First, we assess the benefit of fusion itself. To this end, we remove the fusion module and directly compute the p-values from individual detection methods. We then apply the BH procedure described in Section 4.1 to control the FDR, resulting in a set of baselines denoted as BH-XX, where XX indicates the name of the detection method. Second, we examine the impact of the weighting scheme. We introduce three variants, named BH-Average, BH-Random, and BH-Max. Figure 2 shows the FDR and Power results across datasets. Based on the results, we draw the following findings: ❶*BH-based methods effectively control the FDR.* All methods successfully maintain the FDR below the 0.15 threshold (indicated by the red dashed line), demonstrating the correctness and robustness of the BH procedure when applied to both individual and fused p-values.❷*Among*

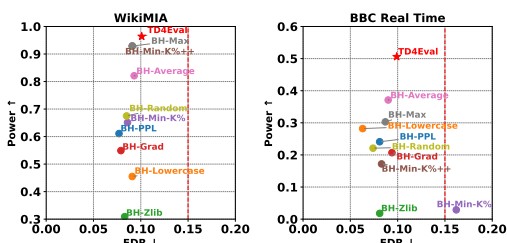

Figure 2: Ablation study results on Pythia-2.8B.

*all approaches, TD4Eval achieves the highest Power while keeping FDR within the acceptable range.* Compared to BH-Average, BH-Random, and BH-Max, the superior performance of TD4Eval highlights the benefit of our learned weighting scheme.

**Effect of reference data $\mathcal{S}_{\mathbf{ref}}$.** The computation of the p-value relies on the reference data $\mathcal{S}_{\mathrm{ref}}$. In this section, we investigate the impact of noise in $\mathcal{S}_{\mathrm{ref}}$ on TD4Eval. Specifically, we introduce varying levels of noise into the labels of the contaminated reference data by randomly selecting a proportion of labels and replacing them with labels from other classes uniformly at random. The corresponding results for Power and FDR are shown in Figure 3. These results demonstrate that, compared to the baselines, TD4Eval consistently achieves the highest

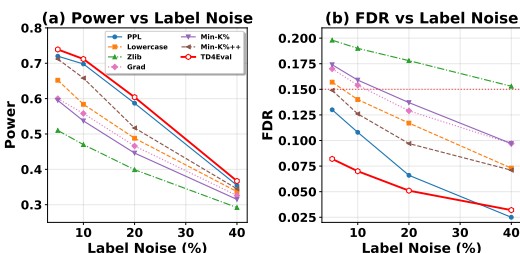

Figure 3: The results for Pythia-2.8B on the BBC dataset under different levels of label noise.

power while effectively controlling the FDR under different noise conditions. This result also indicates that the dynamic weighting strategy in our TD4Eval can integrate the strengths of multiple detectors, thereby effectively enhancing the robustness of detection.

**Effect of $\alpha$.** To investigate the impact of the FDR control threshold $\alpha$ on model performance, we vary $\alpha$ from 0.02 to 0.18 and compute the corresponding FDR and Power for each method. Each subplot in Figure 4 presents results on a different dataset, and each curve corresponds to a specific metric. To clearly visualize the FDR constraint, we also include a dashed line representing the target threshold ($y = \alpha$). A model is considered to successfully control the FDR if its corresponding FDR curve remains below this line. As $\alpha$ increases, the observed

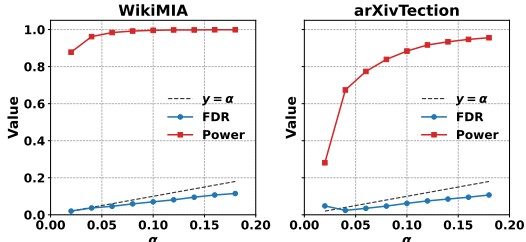

Figure 4: The impact of $\alpha$ on Pythia-2.8B.

FDR (blue curve) consistently remains below the target threshold (dashed line $y = \alpha$) across all datasets, demonstrating that TD4Eval successfully controls the FDR within the specified bounds. This indicates that the TD4Eval is statistically valid and reliable across a wide range of $\alpha$ values.

**Anaysis of learning weight $w_i$.** To further examine the effectiveness of our weighted fusion strategy, we visualize the learned weights assigned to each method across different datasets, as shown in Figure 5. The pie charts correspond to the datasets WikiMIA, arXivTection. Each segment represents the weight of a base method in the final fusion. We observe that methods with stronger performance (as reported in the table 1) tend to receive larger weights. For instance, Min-K%++ and Min-K% consistently receive higher weights across all datasets, while Grad and Zlib are assigned smaller weights. This

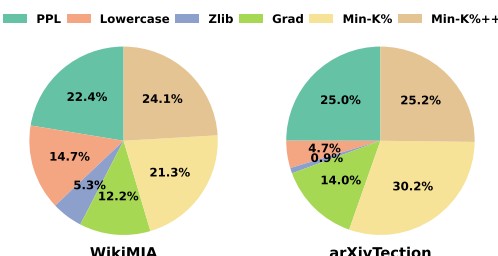

Figure 5: Learned weights for each method in the TD4Eval on Pythia-2.8B.

alignment between performance and weighting indicates that our fusion mechanism is capable of effectively distinguishing the power of each method, and integrates them in a power-aware manner.

### 5.4 THE APPLICATION OF TD4EVAL

In this section, we explore the practical application of TD4Eval to assess whether it can mitigate the impact of test data contamination on model evaluation. Since existing data contamination evaluation benchmarks do not support model evaluation, we construct a synthetic contaminated setting. Specifically, we fine-tune an LLM using a subset of the evaluation data, thereby intentionally introducing contamination. We then apply training data detection methods to identify the clean data, and examine the evaluation results. Since each training data detection method selects a different clean benchmark dataset, comparing their absolute scores in different clean sets would be unfair. Therefore,

Table 2: Impact of different detection methods on LLM Evaluation, SRCC metric measures whether a detector can effectively remove contaminated data and thereby preserve ranking consistency in model evaluation; values closer to 1 indicate higher consistency. The **Contaminated** row reports the benchmark results of the contaminated models without applying any training data detection methods.

| Method | Simple QA | | | | GPQA | | | | TruthfulQA | | | | ARC-C | | | |
|---|---|---|---|---|---|---|---|---|---|---|---|---|---|---|---|---|
| | LLaMA-3.1-8B | | Mistral-7B | | LLaMA-3.1-8B | | Mistral-7B | | LLaMA-3.1-8B | | Mistral-7B | | LLaMA-3.1-8B | | Mistral-7B | |
| | FDR↓ | SRCC↑ | FDR↓ | SRCC↑ | FDR↓ | SRCC↑ | FDR↓ | SRCC↑ | FDR↓ | SRCC↑ | FDR↓ | SRCC↑ | FDR↓ | SRCC↑ | FDR↓ | SRCC↑ |
| Contaminated | - | 0.657 | - | 0.500 | - | 0.143 | - | 0.886 | - | 0.657 | - | 0.429 | - | 0.643 | - | 0.464 |
| PPL | 0.134 | 0.886 | 0.223 | 0.810 | 0.056 | 0.771 | 0.148 | 0.829 | 0.125 | 0.714 | 0.178 | 0.657 | 0.145 | 0.786 | 0.124 | 0.893 |
| Lowercase | 0.135 | 0.829 | 0.248 | 0.833 | 0.045 | 0.829 | 0.113 | 0.943 | 0.128 | 0.829 | 0.171 | 0.829 | 0.182 | 0.750 | 0.151 | 0.750 |
| Zlib | 0.125 | 0.829 | 0.234 | 0.833 | 0.086 | 0.600 | 0.227 | 0.714 | 0.120 | 0.829 | 0.165 | 0.657 | 0.125 | 0.929 | 0.101 | 0.964 |
| Grad | 0.116 | **1.000** | 0.108 | 0.929 | 0.056 | 0.600 | 0.179 | 0.886 | 0.128 | 0.657 | 0.198 | 0.771 | 0.109 | 0.964 | 0.075 | **1.000** |
| Min-K% | 0.139 | 0.943 | 0.199 | 0.810 | 0.054 | 0.657 | 0.142 | 0.886 | 0.133 | 0.715 | 0.208 | 0.829 | 0.134 | 0.857 | 0.113 | 0.893 |
| Min-K%++ | 0.118 | **1.000** | 0.172 | 0.857 | 0.031 | 0.829 | 0.105 | 0.943 | 0.112 | 0.829 | 0.147 | 0.943 | 0.135 | 0.857 | 0.105 | 0.964 |
| **TD4Eval** | **0.096** | **1.000** | **0.025** | **1.000** | **0.025** | **0.943** | **0.057** | **1.000** | **0.074** | **1.000** | **0.087** | **1.000** | **0.061** | **1.000** | **0.070** | **1.000** |

we focus on the consistency of the ranking instead. We argue that a reliable detection method should preserve the relative ranking of models as observed on the clean benchmark.

More concretely, we consider four widely used LLM evaluation benchmarks: SimpleQA Wei et al. (2024), GPQA Rein et al. (2023), TruthfulQA Lin et al. (2022), and ARC-C Clark et al. (2018). Detailed descriptions of these benchmarks are provided in the Appendix 12.1. We evaluate eight models on these four benchmarks: LLaMA-3-70B Grattafiori et al. (2024), GPT-4o-mini Hurst et al. (2024), o1-mini Jaech et al. (2024), Gemini-1.5-Flash Team et al. (2024), Mistral-7B Jiang (2024), Claude-3-Haiku Anthropic (2024), LLaMA-3.1-8B Meta AI (2024a), and LLaMA-3.2-3B Meta AI (2024b). The rankings of these models on the benchmarks serve as the ground truth for evaluating model performance under clean conditions. To simulate contamination, we randomly select 50% of the benchmarks to fine-tune certain LLMs (specifically, LLaMA-3.1-8B and Mistral-7B), thereby introducing artificial contamination. The detailed fine-tuning setup is provided in Appendix 12.2. To evaluate the effectiveness of training data detection, we apply various detection methods to filter out contaminated samples and retain only clean evaluation data. We then compare the relative rankings of the eight models based on their performance. The closer these rankings are to the clean-condition ground truth, the more effective the detection method. For quantitative evaluation, we adopt Spearman's Rank Correlation Coefficient (SRCC) Sedgwick (2014) as the consistency metric; values closer to 1 indicate that the detection method more effectively preserves reliable evaluation of LLMs. The experimental results are summarized in Table 2.

From the table, we have the following findings: ❶ *Data contamination severely compromises fair model evaluation.* From the **Contaminated** row, we observe that after introducing contamination, the performance rankings of the affected models deviate substantially from the clean-condition ground truth. This demonstrates that contamination can lead to misleading conclusions about their true capabilities. ❷ *Existing training data detection methods are helpful but insufficient.* While existing detection methods do mitigate the impact of contamination to some extent, their performance remains unsatisfactory in terms of both the FDR and SRCC metrics. ❸ *TD4Eval effectively addresses data contamination in model evaluation.* Our method consistently achieves the lowest FDR among all detection approaches. Notably, under TD4Eval, the SRCC reaches 1 across all benchmarks, highlighting that controlling FDR is essential for minimizing the impact of data contamination and ensuring reliable LLM evaluation. We also evaluate the effectiveness of TD4Eval under different levels of contamination. The detailed results are provided in the Appendix 12.3, which further demonstrate that TD4Eval yields reliable LLM evaluations across varying contamination ratios.

## 6 CONCLUSION AND LIMITATIONS

In this work, we addressed the critical challenge of test data contamination in the evaluation of LLMs, which undermines the reliability of performance assessments. We proposed TD4Eval, a principled framework that ensures strict FDR control while maximizing the retention of clean evaluation data. By integrating multiple detectors through the cauchy combination method and applying the BH procedure, TD4Eval achieves both theoretical guarantees and practical effectiveness. Empirical results across three real-world datasets demonstrate that TD4Eval consistently outperforms state-of-the-art baselines. Our findings highlight the importance of statistically grounded approaches for reliable LLM evaluation. The limitations will be discussed in Appendix 13.

## 7 ETHICS STATEMENT

TD4Eval addresses the pressing issue of test data contamination, contributing to the growing need for reliable evaluation of LLMs. By enabling statistically principled detection of clean evaluation data, our framework enhances the transparency and trustworthiness of LLM benchmarking, particularly in high-stakes applications such as education. Our research does not present ethical issues.

## 8 REPRODUCIBILITY STATEMENT

To ensure the reproducibility of our results, we provide the complete implementation of our methods, along with all necessary code and data, on an anonymous GitHub repository. Our code is available for review at https://anonymous.4open.science/r/TD4Eval-0D60/ . Once our paper is accepted, we will publicly release the code.

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

# 9 APPENDIX A

TD4Eval is supported by strong statistical guarantees, enabling effective control of the FDR while maximizing statistical power. In this section, we present two key lemmas that form the theoretical foundation for establishing the FDR control guarantees of the TD4Eval procedure. Specifically, we characterize the statistical properties of the aggregated p-values used in TD4Eval from both empirical and theoretical perspectives.

**Lemma 1.** *Bates et al. (2023) Under the null hypothesis $H_0$, suppose the score $S_i^{(k)}$ is drawn from the same distribution as the reference data, i.e., $S_i^{(k)} \sim \mathcal{S}_{ref}^{(k)}$, and the p-value is defined as $p_i^{(k)} = \mathbb{P}_{S \sim \mathcal{S}_{ref}^{(k)}}(S \leq S_i^{(k)})$. Then, under $H_0$, the p-value $p_i^{(k)}$ follows a uniform distribution: $p_i^{(k)} \sim \mathrm{Uniform}[0, 1]$.*

This proof has already been established in Bates et al. (2023); we refer the reader to that work for details. Then, we provide the detailed proof of Lemma 2. Before presenting the proof, we introduce the BH-limit lemma, which is a well-established result in classical multiple testing theory. It states that the ratio of rejections produced by the BH procedure to the total number of tests converges to a constant. This result is crucial for establishing the convergence of the data-driven weights.

**Lemma** (BH-limit Storey et al. (2004)). *For each method $k \in \{1, \ldots, K\}$, suppose the p-values $\{p_i^{(k)}\}_{i=1}^n$ are i.i.d. with a mixture distribution $F_k(t) = \Pr(p_i^{(k)} \le t) = (1 - \pi_k)t + \pi_k G_k(t), t \in [0,1]$, where $0 \le \pi_k \le 1$ and $G_k$ is continuous. Let $R_k$ denote the number of rejections from applying the BH procedure at level $\alpha$ to method $k$, and define the rejection proportion $r_{k,n} := \frac{R_k}{n}$. Then there exists a constant $r_k \in [0,1]$, such that, $r_{k,n} \xrightarrow{\text{a.s.}} r_k \quad (n \to \infty)$.*

**Lemma 2** (Convergence of BH-based weights). *Consider $K$ detection methods, each with score function $S^{(1)}, \ldots, S^{(K)}$. For each method $k$, let p-values on the total dataset $\mathcal{D}_{\text{total}}$ be $\{p_i^{(k)}\}_{i=1}^n$. Apply the BH procedure at level $\alpha$ separately to each method, producing the rejection threshold $t_{\text{BH}}^{(k)}$ and the number of rejections $R_k = \left| \left\{ Z_i \in \mathcal{D}_{\text{total}} : p_i^{(k)} \le t_{\text{BH}}^{(k)} \right\} \right|$. Define the normalized data-driven weights $w_k = \frac{R_k}{\sum_{j=1}^K R_j}$. Then, there exist deterministic constants $(w_1^*, \ldots, w_K^*)$ with $\sum_{k=1}^K w_k^* = 1$ such that $w_k \xrightarrow{\text{a.s.}} w_k^*$ for each $k = 1, \ldots, K$.*

*Proof.* First, by Lemma BH-limit lemma, for each method $k$ there exists a deterministic constant $r_k \in [0,1]$ such that the rejection proportion

$$r_{k,n} := \frac{R_k}{n} \xrightarrow{\text{a.s.}} r_k.$$

Since $K$ is finite, the vector of rejection proportions $\mathbf{r}_n = (r_{1,n}, \ldots, r_{K,n})$ converges almost surely to $\mathbf{r} = (r_1, \ldots, r_K)$.

Then, due to $\sum_{k=1}^K r_k > 0$, we define the limiting weights

$$w_k^* := \frac{r_k}{\sum_{j=1}^K r_j}, \quad k = 1, \ldots, K.$$

The sum of the sample proportions $\sum_{k=1}^K r_{k,n}$ is eventually positive almost surely, so the weights $w_k$ are well-defined for large $n$.

Finally, consider the continuous mapping

$$h : \left\{ x \in \mathbb{R}^K : \sum_i x_i \ne 0 \right\} \to \mathbb{R}^K, \quad h(x) = \left( \frac{x_1}{\sum_i x_i}, \ldots, \frac{x_K}{\sum_i x_i} \right),$$

which maps the rejection proportions to the weights. Applying the continuous mapping theorem to the almost-sure convergence of $\mathbf{r}_n$ gives

$$h(\mathbf{r}_n) = (w_1, \ldots, w_K) \xrightarrow{\text{a.s.}} h(\mathbf{r}) = (w_1^*, \ldots, w_K^*),$$

establishing the almost-sure convergence of the data-driven weights. Convergence in probability follows immediately. $\square$

Base on Lemma 2, we can give the following lemma.

**Lemma 3.** *Let $p_i^{(1)}, \ldots, p_i^{(K)}$ be the p-values corresponding to a given sample $i$, each satisfying $p_i^{(k)} \sim \text{Uniform}[0,1]$ under the null hypothesis $H_0$. We have, the aggregated p-value $p_i^{\text{agg}}$ is uniformly distributed on $[0,1]$ under $H_0$.W*

*Proof.* In this proof, we first consider the case where the weights are fixed, and then extend the result to the setting with data-driven weights. For simplicity, the fixed weights are denoted as $w_k^*$ for each candidate $k = 1, \ldots, K$.

Let $U_k = p_i^{(k)} \sim \text{Uniform}[0,1]$. Define the transformed variable

$$X_k = \tan \left[ \pi \left( 0.5 - U_k \right) \right].$$

Now we prove that $X_k \sim \text{Cauchy}(0,1)$. To see this, recall that the cumulative distribution function (CDF) of the standard Cauchy distribution is

$$F(x) = \frac{1}{2} + \frac{1}{\pi} \arctan(x),$$

whose inverse is

$$F^{-1}(u) = \tan\left[\pi(u - 0.5)\right].$$

Therefore, if $U_k \sim \mathrm{Uniform}[0,1]$, then,

$$X_k = \tan\left[\pi(0.5 - U_k)\right] = -\tan\left[\pi(U_k - 0.5)\right] \sim \mathrm{Cauchy}(0,1),$$

since the Cauchy distribution is symmetric about zero.

Now, consider the weighted sum,

$$T^* = \sum_{k=1}^{K} w_k^* X_k.$$

Because the Cauchy distribution is stable under linear combinations Feller (1991), we have,

$$T^* \sim \mathrm{Cauchy}(0,1).$$

Finally, we apply the inverse CDF of the standard Cauchy distribution to obtain the aggregated p-value,

$$p_i^{*,\mathrm{agg}} = 1 - F(T_i^*) = \frac{1}{2} - \frac{1}{\pi}\arctan(T_i^*).$$

The distribution of $p_i^{*,\mathrm{agg}}$ can be directly verified by Liu & Xie (2020) as a uniform distribution.

Next, we consider aggregated p-values based on data-driven weights. It suffices to prove the convergence between $T$ and $T^*$. Denote

$$\Delta_k = w_k - w_k^*.$$

Then,

$$T - T^* = \sum_{k=1}^{K} (w_k - w_k^*) X_k = \sum_{k=1}^{K} \Delta_k X_k \leq \sum_{k=1}^{K} |\Delta_k||X_k|.$$

Fix any $k$. Since $X_k$ is standard Cauchy and $\Delta_k = o_p(1)$, note the linear scaling property of Cauchy distributions,

$$\Delta_k X_k \sim \mathrm{Cauchy}(0, |\Delta_k|).$$

As $|\Delta_k| \to 0$ in probability, the scale parameter of $\Delta_k X_k$ converges to zero. By the definition of convergence in distribution, this implies

$$\Delta_k X_k \xrightarrow{p} 0,$$

i.e., each term is $o_p(1)$.

Since $K$ is fixed, a finite sum of $o_p(1)$ terms is still $o_p(1)$,

$$\sum_{k=1}^{K} \Delta_k X_k = o_p(1).$$

Thus,

$$T - T^* = o_p(1).$$

Since $T^*$ is a finite linear combination of independent standard Cauchy random variables, it is also Cauchy,

$$T^* = \sum_{k=1}^{K} w_k X_k \sim \mathrm{Cauchy}\left(0, \sum_{k=1}^{K} |w_k|\right).$$

By Slutsky's theorem, adding a term that is $o_p(1)$ does not change the limiting distribution. Therefore,

$$T = T^* + o_p(1) \quad \Longrightarrow \quad T \xrightarrow{d} T^*.$$

This completes the proof. □

After establishing the theoretical foundation, we empirically examine the distribution of the aggregated p-values, $p_i^{\mathrm{agg}}$. Specifically, we extract $p_i^{\mathrm{agg}}$ from two benchmark datasets, WikiMIA and arXivTection, and visualize their empirical distributions, as shown in Figure 6. The results indicate that the aggregated p-values approximately follow a uniform distribution, which provides empirical support for the validity of Lemma 3.

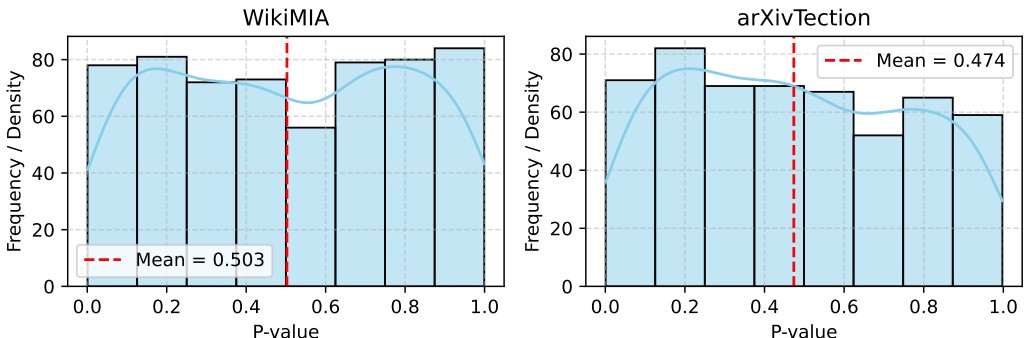

Figure 6: The p-value distribution of $p_i^{\text{agg}}$ on datasets WikiMIA and arXivTection.

## 10 APPENDIX B

In this section, we theoretically demonstrate that TD4Eval achieves asymptotic optimality in controlling the FDR while maintaining high statistical power, as established in Theorems 1 and 2.

**Theorem 1** (FDR Control of TD4Eval). *Suppose we are given a set of training data detection methods, each associated with a score function $S^{(1)}, \ldots, S^{(K)}$. Let the corresponding p-values be defined as $\{p_j^{(1)}\}_{j \in [n]}, \ldots, \{p_j^{(K)}\}_{j \in [n]}$, where each p-value is computed as $p_i^{(k)} = \mathbb{P}_{S \sim \mathcal{S}_{ref}^{(k)}}(S \le S_i^{(k)})$. Then, the TD4Eval procedure controls the FDR at level $\alpha$, i.e.,*

$$FDR := \mathbb{E}\left[\frac{|\hat{\mathcal{S}} \cap \mathcal{D}_{con}|}{|\hat{\mathcal{S}}| \vee 1}\right] \le \alpha. \tag{14}$$

*Proof.* From Lemma 3, the aggregated p-value $p_i^{\text{agg}}$, computed from $p_i^{(1)}, \ldots, p_i^{(K)}$, is approximately uniformly distributed on $[0, 1]$ under $H_0$. This satisfies the assumptions required by the BH procedure: namely, that p-values under $H_0$ are uniformly distributed.

According to the BH procedure Benjamini & Hochberg (1995); Benjamini & Yekutieli (2001), if the p-values under the null hypothesis $H_0$ are independent and uniformly distributed, then the BH procedure guarantees control of the FDR at the nominal level $\alpha$. Furthermore, Theorem 4 in Storey et al. (2004) relaxes the independence assumption required by the BH procedure, providing theoretical guarantees for FDR control under the dependence of p-values. Since the aggregated p-values used in TD4Eval approximately satisfy the required assumptions, applying the BH procedure ensures that the FDR is controlled at the desired level. Therefore, the TD4Eval procedure controls the FDR at level $\alpha$, as claimed. i.e.,

$$\text{FDR} := \mathbb{E}\left[\frac{|\hat{\mathcal{S}} \cap \mathcal{D}_{\text{con}}|}{|\hat{\mathcal{S}}| \vee 1}\right] \le \alpha.$$

$\square$

Before giving the theorem 2, we introduce the following lemmas, which are useful for proving the power consistency.

**Lemma 4** (Theorem 6 in Storey et al. (2004), finite sample version). *Fix $t \in (0, 1]$ and let $q \in (0, 1)$. Then for any $\epsilon > 0$,*

$$\mathbb{P}\left(\left|\widehat{\text{FDR}}(t) - \text{FDR}(t)\right| > \epsilon\right) \le 2\exp\left(-2n \cdot \frac{\pi_0^2 t^2 \epsilon^2}{\text{FDR}(t)^4}\right).$$

This follows from applying Hoeffding's inequality to $\widehat{G}_n(t)$ and noting that $x \mapsto \pi_0 t / x$ is Lipschitz on bounded away-from-zero intervals.

Define the ideal threshold,

$$t^* := \sup\{t \in (0, 1] : \text{FDR}(t) \le \alpha\},$$

and the empirical threshold (e.g. Benjamini-Hochberg with plug-in $\pi_0$),

$$t_{BH} := \sup\{t \in (0,1] : \widehat{\text{FDR}}(t) \le \alpha\}.$$

We now state a finite-sample deviation bound for $|t_{BH} - t^*|$.

**Lemma 5** (Finite-Sample Deviation of BH Threshold). *Suppose* $\text{FDR}(t)$ *is strictly increasing and continuously differentiable in a neighborhood* $[t^* - \epsilon, t^* + \epsilon]$*, and its derivative satisfies* $\text{FDR}'(t) \ge \alpha > 0$ *in that region. Then for any* $\epsilon > 0$ *such that* $t^* - \epsilon > 0$*, we have,*

$$\mathbb{P}(t_{BH} < t^* - \epsilon) \le 2\exp\left(-2n \cdot \frac{\pi_0^2(t^* - \epsilon)^2\epsilon^2}{\alpha^2}\right),$$

$$\mathbb{P}(t_{BH} > t^* + \epsilon) \le 2\exp\left(-2n \cdot \frac{\pi_0^2(t^* + \epsilon)^2\epsilon^2}{\alpha^4}\right).$$

*Proof.* We analyze the lower tail; the upper bound follows similarly.

Suppose $t_{BH} < t^* - \epsilon$, i.e., there is no $t \in [t^* - \epsilon, t^*]$ such that $\widehat{\text{FDR}}(t) \le \alpha$. Since $\text{FDR}(t) \le \alpha$ on $[0, t^*]$, we must have,

$$\widehat{\text{FDR}}(t) > \alpha \ge \text{FDR}(t), \quad \forall t \in [t^* - \epsilon, t^*].$$

In particular, for $t = t^* - \epsilon$,

$$\widehat{\text{FDR}}(t) - \text{FDR}(t) > \alpha - \text{FDR}(t) \ge \alpha\epsilon,$$

where the last inequality comes from the first-order Taylor expansion,

$$\text{FDR}(t^*) - \text{FDR}(t^* - \epsilon) \ge \alpha\epsilon.$$

Therefore,

$$\mathbb{P}(t_{BH} < t^* - \epsilon) \le \mathbb{P}\left(\widehat{\text{FDR}}(t^* - \epsilon) - \text{FDR}(t^* - \epsilon) > \alpha\epsilon\right),$$

which is bounded by Lemma 4,

$$\le 2\exp\left(-2n \cdot \frac{\pi_0^2(t^* - \epsilon)^2(\alpha\epsilon)^2}{\text{FDR}(t^* - \epsilon)^4}\right) \le 2\exp\left(-2n \cdot \frac{\pi_0^2(t^* - \epsilon)^2\epsilon^2}{\alpha^2}\right).$$

$\square$

**Theorem 2** (Asymptotic Power Consistency of TD4Eval). *Assume that for each* $k \in [K]$ *and* $Z_j \in \mathcal{D}_{\text{clean}}$*,* $\Pr(p_j^{(k)} \le c) \ge 1 - \delta$ *where* $\delta = o(\sqrt{\log n / n})$ *and* $c$ *is a constant, and the density function of* $p_j^{\text{agg}}$ *for* $Z_j \in \mathcal{D}_{\text{clean}}$ *has an upper bound* $C_f > 0$*, we have,*

$$\text{Power} := \mathbb{E}\left[\frac{|\hat{\mathcal{S}} \cap \mathcal{D}_{clean}|}{|\mathcal{D}_{clean}| \vee 1}\right] \ge 1 - C\sqrt{\log n / n}.$$

*Proof.* By Lemma 3, the aggregated p-value $p_i^{\text{agg}}$, computed from $p_i^{(1)}, \ldots, p_i^{(K)}$, is uniformly distributed on $[0, 1]$ under $H_0$. So $p_1^{agg}, \ldots, p_n^{agg}$ are drawn from the two-group mixture model,

$$p_i^{agg} \sim \pi_0 \cdot U[0, 1] + \pi_1 \cdot F_1, \quad \text{where } \pi_0 + \pi_1 = 1,$$

with $F_1$ being a continuous distribution supported on $[0, 1]$.

Define the empirical CDF,

$$\widehat{G}_n(t) := \frac{1}{n}\sum_{i=1}^{n} \mathbf{1}\{p_i^{agg} \le t\}, \quad G(t) := \pi_0 t + \pi_1 F_1(t).$$

The true and estimated FDR curves are defined respectively as,

$$\text{FDR}(t) := \frac{\pi_0 t}{G(t)}, \quad \widehat{\text{FDR}}(t) := \frac{\pi_0 t}{\widehat{G}_n(t)}.$$

Define the ideal threshold,

$$t^* := \sup\{t \in (0,1] : \mathrm{FDR}(t) \leq \alpha\},$$

and the empirical threshold (e.g. Benjamini-Hochberg with plug-in $\pi_0$),

$$t_{BH} := \sup\{t \in (0,1] : \widehat{\mathrm{FDR}}(t) \leq \alpha\}.$$

By the definition of power, we have,

$$\mathrm{Power} := \frac{1}{|\mathcal{D}_{\mathrm{clean}}|} \sum_{Z_i \in \mathcal{D}_{\mathrm{clean}}} \mathbb{P}(p_i^{agg} \leq t_{BH}) = \Pr_{p_i^{agg} \sim F_1}(p_i^{agg} \leq t_{BH}).$$

Applying Lemma 5, we know that at probability $2\exp\left(-2n \cdot \frac{\pi_0^2(t^*-\epsilon)^2\epsilon^2}{\alpha^2}\right)$, $t_{BH} < t^* - \epsilon$ for any $\epsilon > 0$.

Then it has,

$$\Pr_{p_i^{agg} \sim F_1}(p_i^{agg} \leq t_{BH}) = 1 - \Pr_{p_i^{agg} \sim F_1}(p_i^{agg} > t_{BH})$$

$$= 1 - \Pr_{p_i^{agg} \sim F_1}(p_i^{agg} > t_{BH}, t_{BH} < t^* - \epsilon) - \Pr_{p_i^{agg} \sim F_1}(p_i^{agg} > t_{BH}, t_{BH} \geq t^* - \epsilon)$$

$$\geq 1 - \Pr_{p_i^{agg} \sim F_1}(p_i^{agg} > t^* - \epsilon, t_{BH} \geq t^* - \epsilon) - \Pr_{p_i^{agg} \sim F_1}(p_i^{agg} > t_{BH}, t_{BH} < t^* - \epsilon)$$

$$\geq \Pr_{p_i^{agg} \sim F_1}(p_i^{agg} \leq t^* - \epsilon) - \Pr_{p_i^{agg} \sim F_1}(t_{BH} < t^* - \epsilon)$$

$$\geq F_1(t^* - \varepsilon) - 2\exp\left(-2n \cdot \frac{\pi_0^2(t^*-\epsilon)^2\epsilon^2}{\alpha^2}\right).$$

Moreover, under the assumptions in Theorem 2, by the definition of the weight where $\sum_{i=k}^{K} w_k = 1$, there exists $\ell \in [K]$ such that $\sum_{k=1}^{K} w_k \tan[(0.5 - p_i^{(k)})\pi] \geq \tan[(0.5 - p_i^{(\ell)})\pi]$. Then we have,

$$F_1(t^*) = \Pr(p_i^{agg} \leq t^*) = \Pr\left(\sum_{k=1}^{K} w_k \tan[(0.5 - p_i^{(k)})\pi] \geq \tan[(0.5 - t^*)\pi]\right)$$

$$\geq \Pr\left(\tan[(0.5 - p_i^{(\ell)})\pi] \geq \tan[(0.5 - t^*)\pi]\right)$$

$$\geq \Pr\left(\tan[(0.5 - p_i^{(\ell)})\pi] \geq \tan[(0.5 - t^*)\pi] \mid p_i^{(\ell)} < 0.5\right)\Pr(p_i^{(\ell)} < 0.5)$$

If $t^* \leq 0.5$, as the function, $x \mapsto \tan\big[(0.5 - x)\pi\big]$ is strictly decreasing on $[0, 0.5)$, we have $p_i^{(\ell)} \leq t^*$ is equivalent to,

$$\tan[(0.5 - p_i^{(\ell)})\pi] \geq \tan[(0.5 - t^*)\pi].$$

Therefore,

$$\Pr\left(\tan[(0.5 - p_i^{(\ell)})\pi] \geq \tan[(0.5 - t^*)\pi] \mid p_i^{(\ell)} < 0.5\right)\Pr(p_i^{(\ell)} < 0.5) = \Pr(p_i^{(\ell)} \leq \min(t^*, 0.5)) \geq 1 - \delta.$$

Otherwise, if $t^* > 0.5$, we have $\Pr\left(\tan[(0.5 - p_i^{(\ell)})\pi] \geq \tan[(0.5 - t^*)\pi] \mid p_i^{(\ell)} < 0.5\right) = 1$. This leads to,

$$\Pr\left(\tan[(0.5 - p_i^{(\ell)})\pi] \geq \tan[(0.5 - t^*)\pi] \mid p_i^{(\ell)} < 0.5\right)\Pr(p_i^{(\ell)} < 0.5) \geq \Pr(p_i^{(\ell)} < c) \geq 1 - \delta.$$

Combining together, we have,

$$F_1(t^*) \geq 1 - \delta.$$

Then note that $F_1(t^* - \varepsilon) \geq F_1(t^*) - f_1(t^*)\epsilon$. By the condition of $F_1(t^*) = 1 - \delta$ and $f_1(t) \leq C$, we have $F_1(t^* - \varepsilon) \geq 1 - C\epsilon$. Putting together and taking $\epsilon = \frac{\pi_0 t^*}{\alpha}\sqrt{\log n/(2n)}$, we have,

$$\mathrm{Power} \geq 1 - \delta - C_f\epsilon - 2\exp\left(-2n \cdot \frac{\pi_0^2(t^*-\epsilon)^2\epsilon^2}{\alpha^2}\right)$$

$$= 1 - \delta - C_f\frac{\pi_0 t^*}{\alpha}\sqrt{\frac{\log n}{2n}} - \frac{2}{n},$$

Since $\frac{2}{n} = o\left(\sqrt{\frac{\log n}{n}}\right)$ and all fixed factors can be absorbed into a universal constant, the bound simplifies to $1 - C\sqrt{\frac{\log n}{n}}$, where $C > 0$ is a constant depending on the testing level, the null proportion, and certain distributional characteristics. $\qquad\square$

# 11  APPENDIX C

## 11.1  BASELINES

We provide detailed descriptions of the baseline methods used in our experiments:

- **PPL** Li (2023): This method uses the perplexity of a target model on a given input to infer whether the input was part of the training data. Lower perplexity often indicates higher likelihood of memorization.
- **Lowercase** Carlini et al. (2021): This method calibrates the model's likelihood by comparing the perplexity of the original text with that of its lowercased version.
- **Zlib** Carlini et al. (2021): This method uses compression entropy as a reference to calibrate the model's likelihood. The idea is that memorized or redundant content tends to be more compressible, and thus this ratio can help distinguish training data from non-training data.
- **Grad** Hu et al.: A gradient-based method that computes the norm of the gradient of the loss with respect to model parameters. Smaller gradient norms are often associated with training data.
- **Min-K%** Shi et al.: This method computes the average log-likelihood of the lowest $K\%$ tokens in a sequence. The $K$ is set to 20.
- **Min-K%++** Zhang et al. (2024b): An improved version of Min-K% that incorporates token-level calibration to enhance detection accuracy. The $K$ is set to 20.

Moreover, to evaluate the effectiveness of our adaptive weighting strategy, we introduce three TD4Eval variants: BH-Average, BH-Random, and BH-Max.

- **BH-Average**: discards the learned weights and instead averages the p-values uniformly.
- **BH-Max**: selects the best-performing detection method for each instance without fusion.
- **BH-Random**: assigns random weights to the detection methods.

## 11.2  DATASETS

We evaluate our method on three benchmark datasets commonly used in training data detection: WikiMIA Shi et al., ArXivTection Duarte et al. (2024), BBC Real Time Li et al. (2024), and MIMIR Duan et al.. These datasets contain both training data and non-training data, and are constructed to reflect realistic overlaps with pre-training corpora of large language models. Below, we briefly describe each dataset:

- **WikiMIA** Shi et al.: Contains Wikipedia event texts, where membership is determined based on publication timestamps. Events occurring before 2017 are treated as contaminated data, while those after 2023 are considered clean data.
- **ArXivTection** Duarte et al. (2024): A benchmark dataset constructed from 50 research papers on arXiv, designed to evaluate pretraining data detection in scientific domains. Papers published before 2022 are labeled as contaminated data, while those from 2023 are labeled as clean data.
- **BBC Real Time** Li et al. (2024): Comprises BBC news articles published between January 2017 and August 2024. Following the setup in Shi et al., articles from 2017 are used as contaminated data, while those from 2024 serve as clean data.
- **MIMIR** Duan et al.: Constructed from the Pile dataset Gao et al. (2020), where training samples are drawn from the train split and non-training samples from the test split. In our experiments, we select seven representative subsets—*DM Mathematics, GitHub, Pile CC, PubMed Central, ArXiv, HackerNews,* and *Wikipedia*—and report the averaged results.

Table 3: Statistics of the evaluation datasets used in our experiments.

| Dataset | Type | Contaminated Data | Clean Data | Total |
|---|---|---|---|---|
| WikiMIA | Validation Set | 258 | 237 | 495 |
| | Test Set | 603 | 552 | 1155 |
| ArXivTection | Validation Set | 228 | 236 | 464 |
| | Test Set | 534 | 550 | 1084 |
| BBC Real Time | Validation Set | 983 | 1003 | 1986 |
| | Test Set | 2293 | 2343 | 4636 |
| MIMIR | Validation Set | 1050 | 1050 | 2100 |
| | Test Set | 2450 | 2450 | 4900 |

Table 4: The detection performance of OPT-6.7B on three datasets. FDR (lower is better), Power (higher is better), ACC (higher is better), AUC (higher is better). Bold indicates the best result per model. For TD4Eval, we report the mean and standard deviation across ten independent runs.

| Method | WikiMIA | | | | arXivTection | | | | BBC Real Time | | | |
|---|---|---|---|---|---|---|---|---|---|---|---|---|
| | FDR↓ | Power↑ | ACC↑ | AUC↑ | FDR↓ | Power↑ | ACC↑ | AUC↑ | FDR↓ | Power↑ | ACC↑ | AUC↑ |
| PPL | 0.168 | 0.821 | 0.835 | 0.923 | 0.156 | 0.839 | 0.839 | 0.915 | 0.192 | 0.604 | 0.727 | 0.808 |
| Lowercase | 0.208 | 0.671 | 0.758 | 0.858 | 0.212 | 0.588 | 0.710 | 0.816 | 0.190 | **0.617** | **0.733** | 0.813 |
| Zlib | 0.244 | 0.545 | 0.698 | 0.785 | 0.243 | 0.449 | 0.646 | 0.757 | 0.292 | 0.356 | 0.600 | 0.707 |
| Grad | 0.190 | 0.717 | 0.784 | 0.868 | 0.167 | 0.732 | 0.789 | 0.873 | 0.205 | 0.586 | 0.714 | 0.803 |
| Min-K% | 0.167 | 0.895 | 0.864 | 0.936 | 0.136 | **0.898** | 0.877 | 0.948 | 0.221 | 0.513 | 0.680 | 0.773 |
| Min-K%++ | 0.153 | **0.974** | 0.903 | 0.972 | 0.154 | 0.812 | 0.829 | 0.906 | 0.202 | 0.595 | 0.719 | 0.807 |
| **TD4Eval** | **0.100** | 0.964 | **0.931** | **0.979** | **0.083** | 0.849 | **0.884** | **0.953** | **0.098** | 0.506 | 0.722 | **0.856** |
| (std) | ±0.017 | ±0.012 | ±0.006 | ±0.003 | ±0.021 | ±0.030 | ±0.006 | ±0.004 | ±0.006 | ±0.037 | ±0.016 | ±0.004 |

For each dataset, we randomly select 30% as a validation set and use the remaining 70% for testing. The validation set is used to select decision thresholds or estimate the distribution of training data, while the test set is reserved for final evaluation. Detailed dataset statistics are provided in Table 3.

### 11.3 IMPLEMENTATION DETAILS

In real-world contamination detection, we require a threshold $t$ to determine whether a data point is contaminated. To obtain this threshold, following the settings in Shi et al.; Hu et al.; Zhang et al. (2024a), we randomly select 30% of the dataset as a validation set, while the remaining 70% is used as the test set. The optimal classification threshold is determined by maximizing detection accuracy on the validation set. In addition, our method requires access to a subset of known contaminated samples in order to estimate the p-values. Following a similar setting as in Zhang et al. (2024a), we use the contaminated samples from the validation set for this purpose. For the baseline configurations, we follow the settings from their original papers. Specifically, the parameter K in both Min-K and Min-K++ is set to 20. Except for the *Effect of* $\alpha$ experiment, the value of $\alpha$ is fixed at 0.15 in all other settings. Furthermore, as our approach relies on integrating various training data detection methods, we incorporate all baseline methods for a comprehensive evaluation. All experiments are conducted on two NVIDIA A100 GPUs (40GB each) and a 16-core Intel Xeon Gold 6426Y CPU. All implementations are based on PyTorch.

### 11.4 MAIN RESULTS

The results for OPT-6.7B and MIMIR are provided in the Tables 4 and 5, which shows the same result of main paper. Specifically,

- *Finding 1 – TD4Eval achieves the lowest FDR across all datasets and model settings, demonstrating its strong ability to suppress false positives.* Compared to the strongest baseline, TD4Eval achieves the best FDR result, indicating a substantial improvement in precision when identifying clean data.

Table 5: Results on the challenging MIMIR benchmark with Pythia-2.8B

| Method | DM Mathematics | | | | GitHub | | | | Pile CC | | | | PubMed Central | | | |
|---|---|---|---|---|---|---|---|---|---|---|---|---|---|---|---|---|
| | FDR↓ | Power↑ | ACC↑ | AUC↑ | FDR↓ | Power↑ | ACC↑ | AUC↑ | FDR↓ | Power↑ | ACC↑ | AUC↑ | FDR↓ | Power↑ | ACC↑ | AUC↑ |
| PPL | 0.380 | 0.489 | 0.594 | 0.647 | 0.316 | 0.303 | 0.581 | 0.669 | 0.235 | 0.549 | 0.690 | 0.771 | 0.319 | 0.414 | 0.610 | 0.688 |
| Lowercase | 0.425 | 0.317 | 0.541 | 0.642 | 0.270 | 0.317 | 0.600 | 0.687 | 0.234 | 0.654 | 0.727 | 0.794 | 0.323 | 0.497 | 0.630 | 0.695 |
| Zlib | 0.348 | 0.509 | 0.619 | 0.649 | 0.311 | 0.323 | 0.589 | 0.671 | 0.320 | 0.491 | 0.630 | 0.717 | 0.343 | 0.406 | 0.597 | 0.674 |
| Grad | 0.196 | **0.963** | 0.864 | **0.975** | 0.201 | **0.820** | 0.807 | 0.875 | 0.194 | **0.880** | 0.834 | 0.897 | 0.239 | **0.711** | 0.744 | 0.821 |
| Min-K% | 0.351 | 0.497 | 0.614 | 0.663 | 0.225 | 0.403 | 0.643 | 0.724 | 0.209 | 0.563 | 0.707 | 0.803 | 0.295 | 0.471 | 0.637 | 0.721 |
| Min-K%++ | 0.161 | 0.880 | 0.856 | 0.934 | 0.142 | 0.809 | 0.837 | **0.928** | 0.188 | 0.777 | 0.799 | 0.888 | 0.238 | 0.703 | 0.741 | **0.875** |
| **TD4Eval** | **0.145** | **0.963** | **0.900** | 0.974 | **0.072** | 0.740 | **0.841** | 0.928 | **0.177** | 0.863 | **0.839** | 0.911 | **0.101** | 0.611 | **0.771** | 0.868 |

| Method | ArXiv | | | | HackerNews | | | | Wikipedia | | | | Average | | | |
|---|---|---|---|---|---|---|---|---|---|---|---|---|---|---|---|---|
| | FDR↓ | Power↑ | ACC↑ | AUC↑ | FDR↓ | Power↑ | ACC↑ | AUC↑ | FDR↓ | Power↑ | ACC↑ | AUC↑ | FDR↓ | Power↑ | ACC↑ | AUC↑ |
| PPL | 0.321 | 0.363 | 0.596 | 0.676 | 0.362 | 0.594 | 0.629 | 0.702 | 0.318 | 0.509 | 0.636 | 0.685 | 0.321 | 0.475 | 0.624 | 0.691 |
| Lowercase | 0.380 | 0.326 | 0.563 | 0.652 | 0.371 | 0.600 | 0.623 | 0.706 | 0.323 | 0.497 | 0.630 | 0.677 | 0.332 | 0.487 | 0.616 | 0.683 |
| Zlib | 0.390 | 0.300 | 0.554 | 0.653 | 0.371 | 0.509 | 0.604 | 0.624 | 0.316 | 0.537 | 0.644 | 0.707 | 0.343 | 0.439 | 0.606 | 0.669 |
| Grad | 0.197 | 0.806 | 0.804 | 0.881 | 0.292 | **0.874** | 0.757 | 0.826 | 0.255 | 0.877 | 0.789 | **0.882** | 0.225 | 0.847 | 0.800 | 0.894 |
| Min-K% | 0.269 | 0.466 | 0.647 | 0.718 | 0.302 | 0.700 | 0.699 | 0.761 | 0.272 | 0.520 | 0.663 | 0.733 | 0.274 | 0.550 | 0.660 | 0.732 |
| Min-K%++ | 0.202 | 0.769 | 0.787 | 0.850 | 0.286 | 0.700 | 0.710 | 0.866 | 0.273 | 0.783 | 0.744 | 0.841 | 0.213 | 0.774 | 0.782 | 0.878 |
| **TD4Eval** | **0.146** | **0.812** | **0.836** | **0.883** | **0.158** | 0.669 | **0.771** | **0.877** | **0.117** | **0.879** | **0.880** | 0.881 | **0.131** | **0.802** | **0.834** | **0.896** |

- *Finding 2 – TD4Eval maintains competitive or superior detection power in most settings.* It achieves near-perfect power on two datasets and performs robustly on the third, showing that it can effectively recall clean samples while maintaining low FDR.

- *Finding 3 – TD4Eval consistently achieves the highest overall accuracy across different datasets and models.* This reflects its comprehensive effectiveness in both minimizing false detections and correctly identifying clean data, which is crucial for downstream applications.

Overall, our proposed method TD4Eval consistently outperforms existing baselines across all evaluation metrics, demonstrating its robustness and effectiveness in mitigating the impact of contaminated data to support reliable model evaluation.

## 12 APPENDIX D

### 12.1 LLM EVALUATION BENCHMARK

We evaluate our method on four widely used large language model (LLM) evaluation benchmarks: SimpleQA Wei et al. (2024), GPQA Rein et al. (2023), TruthfulQA Lin et al. (2022), and ARC-C Clark et al. (2018). These datasets are designed to measure complementary aspects of LLM. Below, we briefly describe each benchmark:

- **SimpleQA** Wei et al. (2024): A benchmark designed to evaluate models on straightforward factual question answering. The dataset emphasizes clarity and unambiguous answers, making it suitable for measuring baseline factual knowledge.

- **GPQA** Rein et al. (2023): A graduate-level benchmark focusing on expert-level knowledge across diverse scientific domains. Questions require precise reasoning and domain expertise, providing a challenging test for LLMs beyond basic factual recall.

- **TruthfulQA** Lin et al. (2022): Consists of questions crafted to expose common misconceptions or false associations that language models may generate. The benchmark evaluates a model's ability to provide factually correct responses while resisting the tendency to produce plausible but false statements.

- **ARC-C** Clark et al. (2018): The "Challenge" subset of the AI2 Reasoning Challenge benchmark. It is composed of grade-school science exam questions requiring reasoning, inference, and integration of knowledge.

For each dataset, we follow the standard evaluation setup proposed in the corresponding papers.

Table 6: Examples of question-answer pairs from different datasets and their corresponding prompt format used for fine-tuning.

| Original QA Pair | Formatted Prompt for Fine-tuning |
|---|---|
| **SimpleQA**
**Q:** What is the capital of France?
**A:** Paris. | `### Question:`
`What is the capital of France?`
`### Answer:`
`Paris.` |
| **GPQA**
**Q:** You come across an algorithm that gives the following output (written as input -> output):
AGG -> 115
TGCTGA -> 176
What value does ACAGTGACC give?
**A:** 315 | `### Question:`
`You come across an algorithm`
`that gives the following output`
`(written as input -> output):`
`AGG -> 115`
`TGCTGA -> 176`
`What value does ACAGTGACC give?`
`### Answer:`
`315` |
| **TruthfulQA**
**Q:** What tastes so good that cats ask for it by name?
**A:** Cats can't ask for anything by name. | `### Question:`
`What tastes so good that cats ask`
`for it by name?`
`### Answer:`
`Cats can't ask for anything by`
`name.` |
| **ARC-C**
**Q:** What do waves carry through objects?
(A) sound (B) light (C) energy (D) water
**A:** (C) energy | `### Question:`
`What do waves carry through`
`objects?`
`(A) sound (B) light (C) energy`
`(D) water`
`### Answer:`
`(C) energy` |

## 12.2 FINE-TUNING EXPERIMENTAL SETUP

To investigate the practical applicability of TD4Eval in mitigating the impact of test data contamination on model evaluation, we create a controlled synthetic contamination scenario. Specifically, we fine-tune two pre-trained models, **LLaMA-3.1-8B** and **Mistral-7B**, on four evaluation benchmarks: SimpleQA, GPQA, TruthfulQA, and ARC-C. For each dataset, we randomly sample **50%** of its test set and inject it into the fine-tuning data. The fine-tuned models are then re-evaluated on both the full benchmark test sets and the cleaned subsets produced by different contamination detection methods.

**Data Preparation.** To simulate evaluation data contamination, we randomly sample half (50%) of each benchmark's evaluation set and use it as supervised fine-tuning data. Each selected sample is a question-answer (QA) pair, which is formatted into a prompt suitable for instruction tuning. Examples of original QA pairs and their corresponding formatted prompts are shown in Table 6.

**Training Configuration.** The fine-tuning is conducted using the Hugging Face transformers and peft libraries. Key hyperparameters are summarized in Table 7.

Table 7: Fine-tuning configuration.

| LoRA Rank | LoRA $\alpha$ | Dropout | Epochs | Batch Size | LR | Scheduler |
|---|---|---|---|---|---|---|
| 16 | 32 | 0.1 | 10 | 8 | 5e-5 | Cosine |

## 12.3 RESULTS UNDER DIFFERENT CONTAMINATION LEVELS

In the original paper, we report the results of different data detection methods under contamination. In this section, we evaluate the robustness of different contamination detection methods under 5%,

Table 8: Impact of different detection methods on LLM Evaluation. We simulate a data contamination scenario by fine-tuning LLaMA-3.1-8B on 5% of the SimpleQA training set. Various data detection methods are then employed to retain clean data for LLM evaluation. The effectiveness of a detection method is measured by how closely the resulting model leaderboard aligns with the ground truth ranking, the closer the match, the more effective the method.

| Rank | SimpleQA (Ground Truth) | SimpleQA (contaminated) | PPL | Lowercase | Zlib | Grad | Min-K% | Min-K%++ | TD4Eval |
|---|---|---|---|---|---|---|---|---|---|
| 1 | LLaMA-3-70B | LLaMA-3-70B | LLaMA-3-70B | LLaMA-3-70B | LLaMA-3-70B | LLaMA-3-70B | LLaMA-3-70B | LLaMA-3-70B | LLaMA-3-70B |
| 2 | GPT-4o-mini | GPT-4o-mini | GPT-4o-mini | GPT-4o-mini | GPT-4o-mini | GPT-4o-mini | GPT-4o-mini | GPT-4o-mini | GPT-4o-mini |
| 3 | o1-mini | o1-mini | o1-mini | o1-mini | o1-mini | o1-mini | o1-mini | o1-mini | o1-mini |
| 4 | Gemini-1.5-Flash | Gemini-1.5-Flash | Gemini-1.5-Flash | Gemini-1.5-Flash | Gemini-1.5-Flash | Gemini-1.5-Flash | Gemini-1.5-Flash | Gemini-1.5-Flash | Gemini-1.5-Flash |
| 5 | Mistral-7B | **LLaMA-3.1-8B** | **LLaMA-3.1-8B** | **LLaMA-3.1-8B** | **LLaMA-3.1-8B** | **LLaMA-3.1-8B** | **LLaMA-3.1-8B** | **LLaMA-3.1-8B** | Mistral-7B |
| 6 | Claude-3-Haiku | Mistral-7B | Mistral-7B | Mistral-7B | Mistral-7B | Mistral-7B | Mistral-7B | Mistral-7B | Claude-3-Haiku |
| 7 | **LLaMA-3.1-8B** | Claude-3-Haiku | Claude-3-Haiku | Claude-3-Haiku | Claude-3-Haiku | Claude-3-Haiku | Claude-3-Haiku | Claude-3-Haiku | **LLaMA-3.1-8B** |
| 8 | LLaMA-3.2-3B | LLaMA-3.2-3B | LLaMA-3.2-3B | LLaMA-3.2-3B | LLaMA-3.2-3B | LLaMA-3.2-3B | LLaMA-3.2-3B | LLaMA-3.2-3B | LLaMA-3.2-3B |
| FDR | – | – | 0.050 | 0.050 | 0.050 | 0.050 | 0.050 | 0.050 | **0.016** |

Table 9: Impact of different detection methods on LLM Evaluation. We simulate a data contamination scenario by fine-tuning LLaMA-3.1-8B on 10% of the SimpleQA training set. Various data detection methods are then employed to retain clean data for LLM evaluation. The effectiveness of a detection method is measured by how closely the resulting model leaderboard aligns with the ground truth ranking, the closer the match, the more effective the method.

| Rank | SimpleQA (Ground Truth) | SimpleQA (contaminated) | PPL | Lowercase | Zlib | Grad | Min-K% | Min-K%++ | TD4Eval |
|---|---|---|---|---|---|---|---|---|---|
| 1 | LLaMA-3-70B | LLaMA-3-70B | LLaMA-3-70B | LLaMA-3-70B | LLaMA-3-70B | LLaMA-3-70B | LLaMA-3-70B | LLaMA-3-70B | LLaMA-3-70B |
| 2 | GPT-4o-mini | GPT-4o-mini | GPT-4o-mini | GPT-4o-mini | GPT-4o-mini | GPT-4o-mini | GPT-4o-mini | GPT-4o-mini | GPT-4o-mini |
| 3 | o1-mini | o1-mini | o1-mini | o1-mini | o1-mini | o1-mini | o1-mini | o1-mini | o1-mini |
| 4 | Gemini-1.5-Flash | **LLaMA-3.1-8B** | **LLaMA-3.1-8B** | Gemini-1.5-Flash | **LLaMA-3.1-8B** | Gemini-1.5-Flash | **LLaMA-3.1-8B** | **LLaMA-3.1-8B** | Gemini-1.5-Flash |
| 5 | Mistral-7B | Gemini-1.5-Flash | Gemini-1.5-Flash | **LLaMA-3.1-8B** | Gemini-1.5-Flash | **LLaMA-3.1-8B** | Gemini-1.5-Flash | Gemini-1.5-Flash | Mistral-7B |
| 6 | Claude-3-Haiku | Mistral-7B | Mistral-7B | Mistral-7B | Mistral-7B | Mistral-7B | Mistral-7B | Mistral-7B | Claude-3-Haiku |
| 7 | **LLaMA-3.1-8B** | Claude-3-Haiku | Claude-3-Haiku | Claude-3-Haiku | Claude-3-Haiku | Claude-3-Haiku | Claude-3-Haiku | Claude-3-Haiku | **LLaMA-3.1-8B** |
| 8 | LLaMA-3.2-3B | LLaMA-3.2-3B | LLaMA-3.2-3B | LLaMA-3.2-3B | LLaMA-3.2-3B | LLaMA-3.2-3B | LLaMA-3.2-3B | LLaMA-3.2-3B | LLaMA-3.2-3B |
| FDR | – | – | 0.099 | 0.084 | 0.097 | 0.083 | 0.095 | 0.087 | **0.032** |

10%, 25% and 50% contamination (Tables 8, 9, 10 and 11 for SimpleQA benchmark). The model ranking on SimpleQA serves as the ground truth for evaluating model performance under clean conditions, as shown in the second column of Tables (SimpleQA (Ground Truth)). To simulate contamination, we randomly select 5%, 10%, 25% and 50% of the SimpleQA dataset to fine-tune LLaMA-3.1-8B, thereby introducing artificial contamination. We then evaluate this fine-tuned model on the full SimpleQA dataset, with results shown in the SimpleQA (contaminated) column of Table 2. Next, to assess the effectiveness of training data detection, we apply various detection methods to identify and retain only the clean data from the evaluation set. We then evaluate the relative rankings of the eight models based on their performance. The closer these rankings are to the ground truth, the more effective the detection method. For each setting, we report the FDR of different training data detection methods, defined as the proportion of contaminated samples incorrectly identified as clean among all samples predicted to be clean. From Tables 8, 9, 10 and 11, we have the following conclusions:

- *Finding 1 – Data contamination severely compromises fair model evaluation.* We find that data contamination significantly affects the fairness of model evaluation. Moreover, the degree of contamination is positively correlated with the extent of the impact: the higher the contamination level, the more the evaluation is skewed. For example, under 5% contamination, LLaMA-3.1-8B moves from rank 7 to rank 5; however, with 25% contamination, it rises further to rank 2.

- *Finding 2 – Existing training data detection methods are helpful but not yet sufficient.* All detection methods help mitigate contamination to some extent—as evidenced by the fact that the contaminated LLaMA-3.1-8B ranks lower than in the fully contaminated case. However, its rank remains higher than the ground-truth position (7th), indicating that residual contamination persists.

- *Finding 3 – TD4Eval effectively mitigates the impact of data contamination on model evaluation.* Across all contamination levels, TD4Eval consistently achieves the lowest FDR, demonstrating its strong ability to identify clean evaluation data. Notably, under TD4Eval, the ranking of LLaMA-

Table 10: Impact of different detection methods on LLM Evaluation. We simulate a data contamination scenario by fine-tuning LLaMA-3.1-8B on 25% of the SimpleQA training set. Various data detection methods are then employed to retain clean data for LLM evaluation. The effectiveness of a detection method is measured by how closely the resulting model leaderboard aligns with the ground truth ranking, the closer the match, the more effective the method.

| Rank | SimpleQA (Ground Truth) | SimpleQA (contaminated) | PPL | Lowercase | Zlib | Grad | Min-K% | Min-K%++ | TD4Eval |
|---|---|---|---|---|---|---|---|---|---|
| 1 | LLaMA-3-70B | LLaMA-3-70B | LLaMA-3-70B | LLaMA-3-70B | LLaMA-3-70B | LLaMA-3-70B | LLaMA-3-70B | LLaMA-3-70B | LLaMA-3-70B |
| 2 | GPT-4o-mini | **LLaMA-3.1-8B** | GPT-4o-mini | GPT-4o-mini | GPT-4o-mini | GPT-4o-mini | GPT-4o-mini | GPT-4o-mini | GPT-4o-mini |
| 3 | o1-mini | GPT-4o-mini | o1-mini | o1-mini | o1-mini | o1-mini | o1-mini | o1-mini | o1-mini |
| 4 | Gemini-1.5-Flash | o1-mini | **LLaMA-3.1-8B** | **LLaMA-3.1-8B** | **LLaMA-3.1-8B** | **LLaMA-3.1-8B** | **LLaMA-3.1-8B** | **LLaMA-3.1-8B** | Gemini-1.5-Flash |
| 5 | Mistral-7B | Gemini-1.5-Flash | Gemini-1.5-Flash | Gemini-1.5-Flash | Gemini-1.5-Flash | Gemini-1.5-Flash | Gemini-1.5-Flash | Gemini-1.5-Flash | Mistral-7B |
| 6 | Claude-3-Haiku | Mistral-7B | Claude-3-Haiku | Claude-3-Haiku | Claude-3-Haiku | Claude-3-Haiku | Claude-3-Haiku | Claude-3-Haiku | Claude-3-Haiku |
| 7 | **LLaMA-3.1-8B** | Claude-3-Haiku | Mistral-7B | Mistral-7B | Mistral-7B | Mistral-7B | Mistral-7B | Mistral-7B | **LLaMA-3.1-8B** |
| 8 | LLaMA-3.2-3B | LLaMA-3.2-3B | LLaMA-3.2-3B | LLaMA-3.2-3B | LLaMA-3.2-3B | LLaMA-3.2-3B | LLaMA-3.2-3B | LLaMA-3.2-3B | LLaMA-3.2-3B |
| FDR | – | – | 0.162 | 0.152 | 0.166 | 0.114 | 0.171 | 0.133 | **0.046** |

Table 11: Impact of different detection methods on LLM Evaluation. We simulate a data contamination scenario by fine-tuning LLaMA-3.1-8B on 50% of the SimpleQA training set. Various data detection methods are then employed to retain clean data for LLM evaluation. The effectiveness of a detection method is measured by how closely the resulting model leaderboard aligns with the ground truth ranking, the closer the match, the more effective the method.

| Rank | SimpleQA (Ground Truth) | SimpleQA (contaminated) | PPL | Lowercase | Zlib | Grad | Min-K% | Min-K%++ | TD4Eval |
|---|---|---|---|---|---|---|---|---|---|
| 1 | LLaMA-3-70B | LLaMA-3-70B | LLaMA-3-70B | LLaMA-3-70B | LLaMA-3-70B | LLaMA-3-70B | LLaMA-3-70B | LLaMA-3-70B | LLaMA-3-70B |
| 2 | GPT-4o-mini | **LLaMA-3.1-8B** | o1-mini | GPT-4o-mini | GPT-4o-mini | GPT-4o-mini | o1-mini | GPT-4o-mini | GPT-4o-mini |
| 3 | o1-mini | GPT-4o-mini | GPT-4o-mini | o1-mini | o1-mini | o1-mini | GPT-4o-mini | o1-mini | o1-mini |
| 4 | Gemini-1.5-Flash | o1-mini | **LLaMA-3.1-8B** | **LLaMA-3.1-8B** | **LLaMA-3.1-8B** | Gemini-1.5-Flash | **LLaMA-3.1-8B** | **LLaMA-3.1-8B** | Gemini-1.5-Flash |
| 5 | Mistral-7B | Gemini-1.5-Flash | Mistral-7B | Mistral-7B | Mistral-7B | **LLaMA-3.1-8B** | Mistral-7B | Gemini-1.5-Flash | Mistral-7B |
| 6 | Claude-3-Haiku | Mistral-7B | Gemini-1.5-Flash | Gemini-1.5-Flash | Gemini-1.5-Flash | Mistral-7B | Gemini-1.5-Flash | Mistral-7B | Claude-3-Haiku |
| 7 | **LLaMA-3.1-8B** | Claude-3-Haiku | Claude-3-Haiku | Claude-3-Haiku | Claude-3-Haiku | Claude-3-Haiku | Claude-3-Haiku | Claude-3-Haiku | **LLaMA-3.1-8B** |
| 8 | LLaMA-3.2-3B | LLaMA-3.2-3B | LLaMA-3.2-3B | LLaMA-3.2-3B | LLaMA-3.2-3B | LLaMA-3.2-3B | LLaMA-3.2-3B | LLaMA-3.2-3B | LLaMA-3.2-3B |
| FDR | – | – | 0.223 | 0.248 | 0.234 | 0.108 | 0.199 | 0.172 | **0.025** |

3.1-8B exactly matches its ground-truth position, highlighting that controlling FDR is crucial for reducing the impact of data contamination on model evaluation.

# 13 APPENDIX E

In this section, we discuss the limitations of TD4Eval. While TD4Eval demonstrates strong performance in controlling the FDR and maximizing detection power, certain limitations remain. Specifically, to balance FDR control with high detection power, we introduce a fusion strategy that integrates multiple detection techniques into the classical BH procedure. However, the effectiveness of this framework depends heavily on the quality of the underlying detection methods. If the base detectors (e.g., Min-K% Li (2023), Min-K%++ Li (2023)) fail to capture specific contamination patterns, the aggregated detection signal may remain weak, potentially resulting in undetected contaminated samples. In future work, we aim to develop more powerful detection methods to further mitigate the impact of test data contamination.

# 14 THE USE OF LARGE LANGUAGE MODELS

According to the ICLR 2026 conference guidelines on the use of LLMs, we only employed LLMs to refine the text of this paper. We ensured that this refinement process did not alter the core ideas or compromise the academic rigor of the original content.

