# OpenReview forum: "Auditing Test Data Contamination with Error Rate Control for Reliable LLM Evaluation"
_ICLR.cc/2026/Conference — ICLR 2026 Conference Withdrawn Submission_

### Official Review · Reviewer_Fgct · 2025-10-27

**Soundness:** 3
**Presentation:** 2
**Contribution:** 2
**Rating:** 4
**Confidence:** 3

**Summary:**

This paper works in the field of test data contamination of LLMs. It proposes a framework called TD4Eval which allows users to specify the desired false discovery rate of contaminted test data and it maximizes the retention of clean evaluation data given that rate. It does so by weighting multiple training data detectors and with those weights, using the BH procedure to control the FDR. They show they identify the contaminated data better than the alternative methods (Table 2) across four datasets and two models each.

**Strengths:**

I think this paper has a lot of potential and the authors have clearly put in a lot of work.  They clearly explain the problem, their frequentist hypothesis testing procedure, a beautiful figure 1 (page 4) and lots of analyses (from page 7 onwards). The idea of taking multple detectors and combining them to produce a new detector which would have some properties, e.g. FDR control, is good. They also establish a clear lower bound on the power of TD4Eval.

**Weaknesses:**

Consider these weaknesses as also questions from my side, as well as my attempts to understand the paper, as not everything was clear to me.

1. The baselines being compared to are also the same detectors that are used as inputs. In other words, this work takes training detectors and looks at whether you can combine them together in some fashion (in this case, Cauchy fusion) to obain better results. To me, this raises a few issues:

(a)  Since they use the same detectors to define both baselines and the input features for their own method, the “30% improvement” is mostly measuring the benefit of tuning and combining existing detectors.
(b) The individual detectors are likely run with default or fixed thresholds, while TD4Eval learns optimal weights and thresholds using partial ground-truth contamination labels.
(c) We do not seem to have a truly independent baseline (e.g. model averaging or simple ensemble vote). This makes it unclear, apart from the statistical properties, why this approach is the one to go with.

2. I'm left uncertain whether this is more theoretical work that aims to establish new underlying theory or a method aimed to be practically useful. If it is the former, I would appreciate greater distinction from what makes this setup distinct so that it wouldn't be a direct appliaction of the BH procedure and showing that its properties hold in this setup (i.e. highlight theoretical contribution). If it's the latter, I would appreciate a much greater emphasis on practicality, such as: how can we use it in practice once we detect data contamination, a larger case study on a big LLM, showing the value of this under settings people are likely to encounter when using this (e.g. train-test domain shift studies).

3.  it's somewhat difficult to read/follow, I wish the authors could structure the paper in a more reader-friendly fashion.

**Questions:**

1. The way you evaluate your model is by artificially contaminating the LLMs (lines 454-457). Is this a standard procedure used in the literature or did you make this up? If standard, can you give some papers that do this already?

2. Table 2 reports the contamined result detection rates. Can you run an ablation where you would show what would happen if you did not fine-tune the models on the data, and fix the same FDR? Do the rankings shift/dont shift?

3. Don't you need to make an assumption that the data distribution is i.i.d. for the training data detectors for you to use them at test time?

4. It seems to me that this method is positioned as solving a practical problem. How would this work in practice? Say, you want to evaluate an LLM. What happens to the LLM evaluation once you've identified the corrupted samples: do you remove them? Weigh them differently? What's the procedure one needs to do to be statistically rigorous and use this method in evaluation?

5. Can you talk a bit about how much time / comp resources it takes to run TD4val?

6. Do I understand correctly that your detectors are also your baselines?

7. When I read this, it seems to me that this is an application of the BH procedure on the test data contamination problem (with an addition of the Cauchy combination). Do you agree with this characterization or is there something distinctly different from simply applying this method to this problem?

8. Could you show what happens when the reference distribution seems to differ more and more from the target distribution (which I imagine is going to be the case in most settings where this is applied)?

9. Can you add confidernce intervals on Table 2 and Table 1?

10. Is the notation for the hypothesis tests different on purpose? Lines 161 and 174.

---

### Official Review · Reviewer_hvhs · 2025-10-29

**Soundness:** 3
**Presentation:** 3
**Contribution:** 2
**Rating:** 2
**Confidence:** 3

**Summary:**

This paper studies the training data detection problem, with a focus on data contamination —i.e., detecting contaminated examples in test sets.

The paper builds on previous works on the topic, including Min-K% and Min-K%++. These methods are referred to as "baseline methods" in the paper.

The paper's contribution is to propose a statistical post-processing method based on scores of different baseline methods. The goal of the post-processing is to control the false discovery rate while maximizing statistical power.

The paper contains a detailed discussion of the proposed method,TD4EVAL, including theoretical guarantees. TD4EVAL is based on relatively advanced statistical techniques, including p-value calculation in the presence of dependencies.

The proposed method is then applied in two different settings. In the first setting, TD4EVAL is used for different datasets (WikiMIA, arXivTection, BBC Real Time) with Pythia-2.8B  and LLaMA-13B . In the second setting, TD4EVAL is applied to detect test questions from Simple QA, GPQA, TruthfulQA, and ARC-C that LLaMA-3.1-8B and Mistral-7B were fine-tuned on. In both settings, the proposed method works well.

The paper also includes several ablation experiments, including comparisons of TD4EVAL with different variants of the Benjamini-Hochberg procedure (Figure 2).

**Strengths:**

- The main contribution of the paper is to introduce advanced statistical techniques for post-processing p-values to the training data detection literature. I don't know the details of the statistical methods on which TD4EVAL is built, but the proposed method makes sense, and the introduction and discussion of TD4EVAL's theoretical properties are rigorous.
- TD4EVAL is an interesting post-processing technique that may be of interest to the ICLR community
- In the experiments, TD4EVAL consistently outperforms the baseline methods.
- The paper has a good overall structure and is very well written.

**Weaknesses:**

**While the paper repeatedly claims to address the problem of data contamination during LLM pre-training, there is no empirical evidence in the paper to support this claim.** To me, this is a very significant concern, because it means that the current version of this paper claims to do something that it just does not do. What this paper *does* show is that the proposed method can detect data contamination that occurred during fine-tuning. However, we know from an increasingly large body of work that membership inference methods that may work for fine-tuning do not work for pre-training.

The most relevant reference here is

- *Zhang et al., Position: Membership Inference Attacks Cannot Prove That a Model was Trained on Your Data, 2025 IEEE Conference on Secure and Trustworthy Machine Learning (SaTML)*

But there are several additional works, including those that perform controlled pre-training experiments with data contamination:

- *Duan et al. , Do Membership Inference Attacks Work on Large Language Models?  COLM 2024*
- *Bordt et al.,  How Much Can We Forget about Data Contamination?, ICML 2025*
- *Kocyigit et al., Overestimation in LLM evaluation: A controlled large-scale study on data contamination’s impact on machine translation., ICML 2025*

At a minimum, for me to raise the score, this paper needs to explain how it overcomes the challenges outlined in Zhang et al. (2025) and Duan et al. (2024).

**The empirical applications that are considered in this paper are not very challenging.** For example, almost all methods, including the baselines, achieve an AUC of >0.9 in Table 1. I don't know what exactly the reasons are for this, but when I compare, for example, with Figure 1 in the Min-k ++ paper, the empirical application there looks much more challenging.

The fact that this application is relatively easy, even for the baseline methods, makes it hard to judge the benefits of TD4EVAL. Does there exist a setting similar to Table 1 where the baseline methods do not perform very well, and TD4EVAL outperforms them significantly?

In addition, the methods BH-XX from Section 5.3 should be included in the main results Table 1 and Table 2.

**In my view, the paper conflates the training data detection problem with the data contamination detection problem.** I actually agree with the framing of the problem in this paper to a large degree, for example, when the authors note that

*the problem of data contamination can, to some
extent, be viewed as a specific instance of Membership Inference Attacks [...] which aim to determine whether a given data point was part of a model’s training
set.*

However, I also believe that determining whether a model was trained on arXiv or Wikipedia is significantly easier than identifying the subset of test-set questions that should be judged as contaminated -- unless the experimental setup is artificially biased to favor the detection method.

**Questions:**

**Comment:** With respect to the validity of your results in the pre-training setting, my impression is that the critical assumption is access to a set S_ref that has the same distribution as the relevant data, and where we know that it was contaminated (for example, in the statement of Lemma 1). My understanding of Zhang et al. (2025) and Duan et al. (2024) is that the problem lies in the fact that we never have access to such an S_ref in the pre-training scenario.

**Justification for final score:**

This paper proposes to employ advanced statistical methods to improve the statistical power of training data detection methods while maintaining FDR control. The derivation of the method itself is sound, and the idea to apply advanced variants of Benjamini-Hochberg for training data detection is interesting. However, the paper does not provide strong empirical evidence that the proposed method outperforms reasonable baselines (BH-XXX) in realistic application scenarios. In addition, the paper repeatedly claims that the proposed method can address data contamination in LLMs, a claim for which the paper does not provide evidence; the literature suggests that this is a highly challenging problem. Because of these concerns, I conclude that the currently submitted version of this paper is not good enough for acceptance at ICLR. I want to stress that the submitted research is fundamentally interesting, and the paper overall is well-executed.

---

### Official Review · Reviewer_9dgf · 2025-10-30

**Soundness:** 3
**Presentation:** 2
**Contribution:** 3
**Rating:** 4
**Confidence:** 4

**Summary:**

The paper presents a framework for detecting contamination in training data that provides good controllability in terms of false discovery rate as well as good detection power. The experiments evaluate on a number of datasets and models and also provide a minimal leaderboard example that shows the value of the method.

**Strengths:**

The problem of data contamination is important and timely.

The method addresses the right challenges present in practice and makes realistic assumptions about what is important.

Experiments are extensive and diverse

**Weaknesses:**

The presentation of the paper can be improved. First, the FDR terminology can be set in a TP / FP regime or terminology which is generally more aligned with the ML terminology but of course this is a matter of taste and background. Second, the method as is does have a series of steps, many of which have tunable parameters. Although the work does discuss the impact of each parameter, a figure or table summary of each step along with assumptions and knobs would help the reader understand the tradeoffs in each step.

The paper does not study many RL trained models yet except o1-mini. One could consider adding perhaps GPT-OSS or one of the RL-trained Qwen models for this purpose as contamination during RL training is a completely different phenomena.

For a lot of the work in this topic, it has become a standard using Pythia but it would have been useful to see ablation studies on other oss models too that also have a lot of transparency around data such as Olmo 2 and Nemotron.

**Questions:**

- How do the authors explain the fact that for SimpleQA LLaMA-3-70B is at the top of the leaderboard? The model was released in April 24 but the benchmark was released in Oct 24.

---

### Official Review · Reviewer_FTcV · 2025-10-31

**Soundness:** 2
**Presentation:** 3
**Contribution:** 2
**Rating:** 2
**Confidence:** 4

**Summary:**

This paper addresses the critical problem of test data contamination, which undermines the reliability of LLM evaluations. The authors propose TD4Eval, a novel statistical framework designed to audit benchmark datasets and identify "clean" (unseen) data for reliable evaluation.

**Strengths:**

- The paper's primary strength is its novel and highly significant reframing of the data contamination problem. Moving beyond simple classification accuracy to a principled statistical framework focused on False Discovery Rate (FDR) control is a major conceptual contribution. This objective is far more aligned with the practical needs of reliable evaluation, where the main concern is limiting the number of false positives (i.e., contaminated samples) in the resulting "clean" evaluation set.

- The proposed TD4Eval framework is theoretically elegant and statistically well-grounded. The combination of an adaptive, data-driven weighting strategy , a weighted Cauchy fusion for p-value combination , and the standard Benjamini-Hochberg procedure  creates a complete, end-to-end, principled method.

**Weaknesses:**

Despite the novelty of its problem formulation, the paper's proposed solution (TD4Eval) suffers from several critical weaknesses in its assumptions, validation, and practical utility.

1. The entire statistical framework is built on the ability to compute accurate p-values. This, in turn, depends entirely on having access to a representative reference distribution of true contaminated data, $S_{ref}$. The paper's proposed solution—approximating $S_{ref}$ using "commonly used pretraining sources" —is a critical and likely fatal flaw. In a real-world scenario, it is impossible to know the true distribution of contaminated data. The actual contamination in a benchmark might come from a niche source (e.g., a specific textbook, GitHub repositories) whose score distribution looks nothing like a generic Wikipedia dump. If $S_{ref}$ is a poor approximation, the p-values are invalid, and the theoretical FDR control guarantee of the BH procedure no longer holds. The paper fails to test the sensitivity of its method to this crucial assumption.

2. The paper fails to validate its method against the most relevant forms of contamination.
**(a)** The main application experiment in Section 5.4 simulates contamination by fine-tuning an LLM on benchmark data. This is a form of intense, deliberate overfitting and is a much easier detection problem than the one the paper claims to solve: detecting data inadvertently seen once or twice during pre-training within a massive, multi-trillion-token corpus. The near-perfect results (SRCC=1.0)  are likely an artifact of this unrealistic simulation and do not prove the method's utility for the more subtle pre-training case.
**(b)** The framework inherits the weaknesses of its base detectors (PPL, Min-K%, etc.), which primarily capture literal string overlap . The paper provides no evidence or discussion on detecting semantic contamination (e.g., paraphrased questions, translated benchmarks, or problems whose concepts were seen, even if the text was not). This is a major blind spot.

3. The TD4Eval framework requires running $K$ different base detectors (where $K=6$ in this paper) for every single sample in the dataset being audited. Many of these detectors (e.g., Grad) are already computationally expensive. This K-fold amplification of cost makes the method practically infeasible for auditing large-scale, real-world benchmarks that may contain tens or hundreds of thousands of samples.

4. The adaptive weighting strategy assigns weights based on the number of rejections $R_k$. This inherently rewards detectors that are more "aggressive" (i.e., reject more hypotheses). While this maximizes power, it may also make the system less stable. A detector that is "lucky" on a validation set, or one that barely meets the FDR threshold (e.g., 0.149 for an $\alpha=0.15$ target), could be assigned a high weight, pushing the entire fused system closer to the edge of failure. A more conservative, variance-aware weighting scheme might be more robust.

**Questions:**

1. How sensitive is TD4Eval's FDR control to a mismatch in the reference distribution? Could you provide an experiment where $S_{ref}$ is generated from one domain (e.g., Wikipedia) but the actual contamination comes from another (e.g., arXiv)? If the p-values are wrong, the entire theoretical guarantee collapses. This seems to be the most critical unverified assumption.

2. Can you provide evidence that TD4Eval works for "pre-training" contamination, not just "fine-tuning" contamination? For example, could you train a smaller model from scratch and guarantee a single pass over a known contaminated sample, and then show that your method can detect it? This would be a much more convincing validation of the method's utility.

3. Do you have any evidence that your framework (or any of its base detectors) can identify non-literal, semantic contamination (e.g., a paraphrased version of a test question)? If not, should this be stated as a **major limitation**, as it's a key way benchmark creators are currently trying to avoid contamination?

4. Could you provide a concrete analysis of the computational overhead? For example, how many GPU-hours does it take to audit the full WikiMIA test set ($n=1155$) using TD4Eval versus using only the best single baseline (Min-K%++)? This is essential for understanding if the method is practical.

5. Your weighting scheme rewards power (high $R_k$). Did you consider an alternative scheme that also rewards stability or robustness (e.g., a detector that achieves a much lower FDR than the target $\alpha$)? It seems the current strategy might favor aggressive detectors and risk instability.

---

### Note · Authors · 2026-01-05

I have read and agree with the venue's withdrawal policy on behalf of myself and my co-authors.